# QUERY-BASED KNOWLEDGE TRANSFER FOR HETEROGENEOUS LEARNING ENVIRONMENTS

**Norah Alballa**♻, **Wenxuan Zhang**♻, **Ziquan Liu**♛, **Ahmed M. Abdelmoniem**♛,
**Mohamed Elhoseiny**♻, **Marco Canini**♻
♻ KAUST          ♛ Queen Mary University of London

## ABSTRACT

Decentralized collaborative learning under data heterogeneity and privacy constraints has rapidly advanced. However, existing solutions like federated learning, ensembles, and transfer learning, often fail to adequately serve the unique needs of clients, especially when local data representation is limited. To address this issue, we propose a novel framework called Query-based Knowledge Transfer (QKT) that enables tailored knowledge acquisition to fulfill specific client needs without direct data exchange. QKT employs a data-free masking strategy to facilitate communication-efficient query-focused knowledge transfer while refining task-specific parameters to mitigate knowledge interference and forgetting. Our experiments, conducted on both standard and clinical benchmarks, show that QKT significantly outperforms existing collaborative learning methods by an average of 20.91% points in single-class query settings and an average of 14.32% points in multi-class query scenarios. Further analysis and ablation studies reveal that QKT effectively balances the learning of new and existing knowledge, showing strong potential for its application in decentralized learning.

## 1 INTRODUCTION

Collaborative model training across distributed sources has made significant progress, where data aggregation to update a central model has become a widely adopted approach. However, the rapid proliferation of Internet of Things (IoT) devices and the increasingly stringent data privacy regulations have highlighted the need for a decentralized machine learning framework. This framework allows models to be trained locally on devices or within organizations and encourages knowledge transfer between models in the network of clients without exchanging raw data. Despite its potential, the decentralized paradigm faces substantial challenges, particularly in addressing the diverse needs of devices and clients in heterogeneous environments.

In heterogeneous environments, each client may have vastly different local data distributions, resulting in diverse query objectives that might be out of the local distribution but relevant to other clients. For instance, in medical diagnostics, models may be required to detect rare or emerging diseases that are underrepresented locally, necessitating the ability to generalize from similar conditions observed in other regions or populations. Similarly, in fraud detection, the constantly evolving nature of fraudulent activities means that new tactics may not yet be captured in the historical data of certain clients. Consequently, it is helpful for models to rapidly learn from fraud patterns detected elsewhere to remain effective.

Previous work has offered valuable solutions to this challenge, but each comes with its own limitations. Collaborative methods like Federated Learning (FL) (McMahan et al., 2017) aggregate knowledge across clients but often struggle to adapt models to the specific needs of individual clients. Personalized FL and clustered FL (Tan et al., 2022; Ma et al., 2022) provide some level of customization by tailoring the learning process to each client's data distribution or by grouping clients of similar characteristics. However, these methods

Table 1: Comparison of Existing Approaches and QKT in Collaborative Learning

| Approach | Data Privacy | Low Comm. Overhead | Customized Models | Knowledge Retention | Customized Objective |
|---|---|---|---|---|---|
| Traditional FL | ✓ | ✗ | ✗ | ✗ | ✗ |
| Personalized FL | ✓ | ✗ | ✓ | ✓ | ✗ |
| KD | ✓ | ✓ | ✗ | ✗ | ✗ |
| Ensemble & Matching (e.g., CLUE) | ✓ | ✓ | ✓ | ✗ | ✗ |
| Continual Learning (e.g., replay) | ✗ | ✓ | ✓ | ✓ | ✗ |
| QKT (Our Method) | ✓ | ✓ | ✓ | ✓ | ✓ |

overlook the scenarios where critical knowledge might be limited or absent in the local data (Fallah et al., 2020; Li et al., 2020a). Moreover, existing FL methods often fail to address the communication overhead, which is crucial when models need to adapt frequently to diverse needs. While approaches such as single-round FL (Yurochkin et al., 2019), ensemble (Daga et al., 2023), knowledge distillation (KD) (Hinton et al., 2015), and replay (Chaudhry et al., 2019) offer potential communication-efficient solutions, they encounter challenges among complex model structures (Wang et al., 2020b), heterogeneous data distributions, privacy constraints, knowledge interference, and catastrophic forgetting (Alballa & Canini, 2024). Table 1 shows the pros and cons of the solutions when addressing the challenge.

These challenges points to the need for an approach that effectively tailors the learning to clients' specific new needs with data privacy and low communication overhead without compromising the existing needs.

In this work, we explore the existing collaborative learning methods in a decentralized environment and observe that they often struggle with maintaining the ability of existing knowledge when learning the queried knowledge. To address this, we propose Query-based Knowledge Transfer (QKT) to enhance a local model, referred to as the student model, by distilling knowledge from other models in the network, referred to as teacher models. Specifically, we generate a synthetic dataset to create a mask that filters out teacher models and parameters associated with irrelevant knowledge during the distillation process. We then introduce a separate phase dedicated to refining the classification head of the student model. This staged distillation process ensures that new knowledge is incorporated effectively without disrupting previously acquired information, thereby mitigating issues such as knowledge interference and forgetting.

We demonstrate the effectiveness of QKT in enhancing model performance on tasks with limited or no local data representation. The results of various standard and clinical benchmarks show that QKT significantly outperforms existing collaborative learning methods. QKT improves learning effectiveness while mitigating key issues in collaborative environments, such as knowledge interference and catastrophic forgetting. Additionally, QKT reduces communication overhead and preserves flexibility in model architecture, allowing teacher models to function as black boxes, thus eliminating the need for model architecture uniformity across all devices. Our contribution can be summarized as:

- We explore decentralized learning and identify the limitations of existing collaborative learning methods in addressing the unique requirements of clients with limited or no local data representation.
- We propose Query-based Knowledge Transfer (QKT) that focuses on query-based learning through two distinct strategies: *Query-Focused Learning* and *Classification Head Refinement*. QKT does not require direct data exchange and minimizes communication overhead.
- We evaluate QKT on various standard and clinical datasets with multiple heterogeneous distributions and a range of scenarios of single query and multiple queries with limited or no local data representation. QKT outperforms existing collaborative learning methods by an average of 20.91% points in single-query scenarios and an average of 14.32% points in multi-query scenarios.

## 2 RELATED WORK

**Federated learning.** FL methods focus on aggregating knowledge across clients to optimize model performance. Approaches like FedProx (Li et al., 2020b) and MOON (Li et al., 2021b) employ regularization techniques to stabilize learning, but often fall short in addressing the wide range of client-specific require-

ments. Recent advances in personalized FL (pFL) tailor learning to each client's data distribution (Qin et al., 2023; Yu et al., 2022; Huang et al., 2021; T. Dinh et al., 2020; Li et al., 2021c; Luo & Wu, 2022; Kharrat et al., 2025), and clustered FL groups clients by data similarities to optimize shared models (Li et al., 2021a; 2022; Ghosh et al., 2020; Briggs et al., 2020; Sattler et al., 2020). However, these approaches often fail when clients require insights that are under-represented or unseen in their data, limiting their applicability in diverse scenarios. Additionally, FL methods face challenges related to communication efficiency.

**Communication-efficient knowledge transfer.** Multiple methods in both FL and other fields try to transmit teacher models in a single round to the learner clients and learn from these models. Ensemble methods apply techniques like bagging, boosting, and stacking (Dietterich, 2000) to combine multiple models and aggregate their predictions. In federated and collaborative learning, ensembles have been employed to mitigate model heterogeneity (Shi et al., 2024). However, they add computational and storage overhead while not fully addressing the challenges of knowledge transfer in heterogeneous data environments. Matching methods, like PFNM (Yurochkin et al., 2019), employ a Bayesian non-parametric model to match and merge local models into a global one. While it reduces communication rounds, storage, and computation, it struggles with heterogeneity in data distributions and complex model structures (Wang et al., 2020a), leading to suboptimal performance in diverse environments. CLUE (Daga et al., 2023) introduces a dynamic approach to knowledge transfer by identifying significant parameters from teacher models and integrating them into student models using a multi-modal boosting technique. However, replacing or averaging significant parameters from models trained on different data can degrade performance and cause substantial forgetting in heterogeneous data settings. KD (Hinton et al., 2015) is an efficient method to transfer knowledge from a teacher model to a student model in a single communication round. However, it introduces challenges such as knowledge interference and forgetting, particularly in heterogeneous environments. These challenges and strategies to address them will be discussed in detail in the next section. Further discussion of other methods, including additional FL baselines and replay-based continual learning is provided in Appendix A.1.

## 3 QUERY-BASED KNOWLEDGE TRANSFER (QKT)

### 3.1 PROBLEM FORMULATION

In this work, we focus on image classification within a collaborative learning environment. Let $x \in \mathcal{X} \subseteq \mathbb{R}^m$ be images and $y \in \mathcal{Y} \subseteq [C]$ be labels, where $[C]$ is integers from 0 to $C - 1$. $f_\theta : \mathcal{X} \to \mathcal{P}_C$ is a model parameterized by $\theta$ that maps an input image to $C$-dimension probability space representing the predictive probability for each class. It can be decomposed into two parts, a feature extractor ($g_\nu$) and the classification head ($h_\mu$), *i.e.*, $f_\theta = h_\mu \circ g_\nu$. We assume there are $L$ clients, each owning a private dataset $\mathcal{D}_i = \{(x, y) | x \in \mathcal{X}_i, y \in \mathcal{Y}_i \subset \mathcal{Y}\}$. Reflective of the real-world environments, these datasets could vary in the volume of data and may also exhibit significant differences in their distribution characteristics. Each client $i \in [L]$ creates a model $f_{\theta_i}$ optimized on its local data, *i.e.*, $\theta_i = \text{argmin} \sum_{(x,y) \in \mathcal{D}_i} \mathcal{L}(f_{\theta_i}(x), y)$, where $\mathcal{L}$ is a supervised loss. After the local training, clients are limited to sharing only the model weights, with the exchange of raw data strictly prohibited.

We study the query-based knowledge transfer in a decentralized environment. For a specific student client $s \in [L]$, the objective is to enhance the performance of model $f_{\theta_s}$ on a set of classes, referred to as the query class(es) $\mathcal{Q}$, leveraging the model weights shared by other clients in the environment, denoted as teacher clients $f_{\theta_t}, t \in [L]$. That is, the objective is to train a student model $f_{\theta_s}$ with local data $\mathcal{D}_s$ and teacher models $f_{\theta_t}$ such that the empirical risk of the query classes $\mathcal{Q}$ is reduced. The selection of the query class(es) $\mathcal{Q}$ are based on the under-representation or absence of the student client's local data in general but can also be other triggers such as performance-based or proactive queries.

### 3.2 CHALLENGES OF KNOWLEDGE DISTILLATION

Knowledge Distillation (KD) is a widely used method to facilitate learning between models without direct access to raw data (Seo et al., 2022; Qin et al., 2024). In KD, the Kullback-Leibler (KL) divergence is

Figure 1: Extraneous knowledge, whether from additional classes or non-proficient teachers, can interfere with the query and local classes, leading to unsatisfactory performance for the query classes.

minimized between the teacher model's predictions $p_t(x) = f_{\theta_t}(x)$ and the student model's predictions $p_s(x) = f_{\theta_s}(x)$. To maintain the knowledge that the student model has learned from its local data, it is often combined with the cross-entropy loss on the local dataset $\mathcal{D}_s$. The standard loss for naive KD is:

$$\mathcal{L}_{\text{KD}}(f_{\theta_s}) = \frac{1}{|\mathcal{D}_s|} \sum_{(x,y) \in \mathcal{D}_s} \{ \text{CE}(p_s(x), y) + \alpha \sum_{t \in \mathcal{T}} \text{KL}(p_t(x), p_s(x)) \}, \tag{1}$$

where $\text{KL}(\cdot, \cdot)$ is the KL divergence, $\mathcal{D}_s$ is the dataset of the student client with size $|\mathcal{D}_s|$, $\text{CE}(\cdot, \cdot)$ is the cross-entropy loss, and $\alpha$ is a weight to balance the two terms.

Through highlights of experiments, we now reveal several key observations that inform the enhancement of learning outcomes in our QKT framework, including improving knowledge inference and mitigating forgetting problems. We utilize the naive KD in our decentralized learning environment to transfer the knowledge for the query classes from the teacher models to the student model and visualize the experiments of models trained on CIFAR10 (Krizhevsky, 2009) with pathological data distribution. The experimental setup is detailed in Section 4.1.

**Irrelevant knowledge interference.** Extraneous irrelevant knowledge can disrupt the learning process by interfering with both the target and existing knowledge. This irrelevant knowledge might come from teachers who are less proficient in the target (i.e., query) classes or from additional classes that do not align with the student's objectives. As shown in Figure 1, when naive KD is applied to the student model, the student model acquires the query knowledge (orange bars), as well as the irrelevant knowledge (gray bars), from the teacher models. This extraneous knowledge negatively impacts the preservation of previously learned information (illustrated with the drop in blue bars), and the effective learning of the target classes. To mitigate these issues, it is essential to filter out unhelpful teachers and irrelevant classes to better fulfill the learner's query. The challenge, however, is to address the following questions:

(1) How can we effectively estimate the knowledge of each teacher model and mask out irrelevant knowledge without access to sensitive or unknown statistical information? (2) How can we modify the KD loss to filter out irrelevant knowledge and focus effectively on the required knowledge?

**Critical role of task-related parameters.** Figure 2, the scatter plots represent the features obtained through Principal Component Analysis (PCA) of the extracted features, denoted as $g_\nu(x)$, from samples belonging to both query and local classes. The background colors in these plots represent the predicted decision boundaries. These decision boundaries are derived from the model's classifier, showing which regions of the feature space are assigned to which class. In the bottom row, heatmaps display each model's class accuracy. These heatmaps quantify the decision boundaries' effectiveness, revealing how accurately the defined regions in the feature space correspond to correct classifications. The Local Training model achieves well-defined boundaries for local classes but fails to generalize to the query class Q. Naive KD improves decision boundaries but struggles to learn the query class Q. In QKT Phase 1, using the query-focused learning we will introduce later, there is a notable improvement in the accuracy of class Q, although the performance of certain local classes, like L2, is suboptimal. QKT Phase 2, in which only the classification head is refined while the feature extractor $g_\nu$ from Phase 1 is frozen, results in a notable accuracy increase particularly for previously under-performing classes such as L2. Our observations highlight the pivotal role of task-related parameters, particularly the classification head in classification tasks. It translates features into specific, actionable class information required by the classification task.

As such, focusing on classification head refinement proves essential in improving learning outcomes. Note that this finding is consistent with existing research (Kumar et al., 2022) that demonstrates the importance

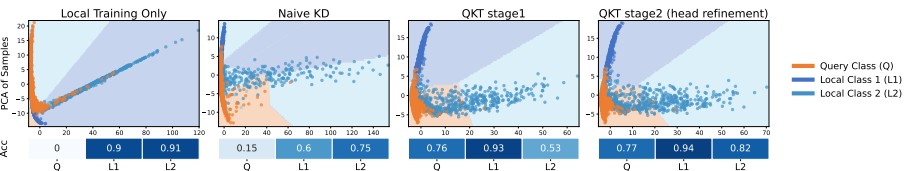

Figure 2: Decision boundaries and class accuracy of Local Training, Naive KD, QKT Phase 1, and QKT Phase 2. Refining the classification head in QKT Phase 2 markedly improves the performance.

of linear probing (only fine-tuning the head) when the model learns an out-of-distribution task. Our work generalizes this principle to the collaborative learning setting and proposes a head replacement strategy to retain previously learned knowledge.

### 3.3 PROPOSED METHOD

**Overview.** We propose Query-based Knowledge Transfer (QKT) to address the challenges of decentralized learning with two key techniques: *Query-Focused Learning* and *Classification Head Refinement*. The learning process occurs in two distinct phases to optimize the model's performance: the first phase focuses on enhancing the feature extractor, while the second phase refines the classification head to ensure effective integration of new knowledge while preserving previously acquired information, as outlined in Figure 3.

**Phase 1: Feature extractor enhancement.** In the first phase, we apply *Query-Focused Learning* to mitigate the interference of irrelevant knowledge by filtering out irrelevant teachers and classes. To achieve this, we apply synthetic data to estimate the relevance of the teachers to the student's query to obtain masks. The teacher models are pre-trained on their local data distributions; they tend to exhibit overconfidence in the classes they were originally trained on (Guo et al., 2017). This overconfidence is reflected in their high probabilities of random input to the learned classes, as we visualize in Figure 5. To this end, a batch of $B$ input samples $x' \sim \mathcal{N}(0, 1) \in \mathbb{R}^m$ generated from a Gaussian distribution is fed into the teacher models to compute class probabilities. If the average prediction of a class $j \in [C]$ surpasses a pre-defined threshold,

$$\mathcal{T}_{\mathcal{Q}} = \{t \mid \tfrac{1}{B} \sum_{x' \sim \mathcal{N}(0,1)} f_{\theta_t}(x')[q] \geq \tau, \exists q \in \mathcal{Q}\}, \tag{2}$$

where $f_{\theta_t}(x')[q]$ represents the probability of class $q$ predicted by teacher $t$ using the synthetic data $x'$, and $\tau$ is the threshold. In practice, the threshold $\tau$ can be easily set as a small number as the irrelevant classes have almost zero probability due to the over-confidence we explained above and we analyze in Section 4.3.

For the selected teacher models, we assign a value of $\lambda$ at positions corresponding to the query classes, 1 at positions corresponding to the student's local classes (to mitigate forgetting), and 0 elsewhere. The mask $\mathbf{M_t} \in \mathbb{R}^C$ is defined as follows:

$$\forall j \in [C], \quad \mathbf{M_t}[j] = \begin{cases} \lambda & \text{if } j \text{ is a query class} \\ 1 & \text{if } j \text{ is present in student's data} , \\ 0 & \text{otherwise} \end{cases} \tag{3}$$

where $\lambda$ controls the emphasis on learning the query class. We discuss the impact of $\lambda$ in Section 4.3. The refined Query-based Knowledge Distillation (QKD) loss is then defined as:

$$\mathcal{L}_{\text{QKD}}(f_{\theta_s}) = \tfrac{1}{|\mathcal{D}_s|} \sum_{(x,y) \in \mathcal{D}_s} \{ \text{CE}(f_{\theta_s}(x), y) + \alpha \sum_{t \in \mathcal{T}_{\mathcal{Q}}} \langle \mathbf{M_t}, \mathbf{KL}(p_t, p_s) \rangle \}, \tag{4}$$

where $\text{CE}(\cdot, \cdot)$ is the cross-entropy loss, and $p_s = f_{\theta_s}(x)$, $p_t = f_{\theta_t}(x)$ represent the student and teacher model predictions, respectively. The term $\mathbf{KL}$ is a vector of element-wise cross-entropy between the teacher and student predictions, specifically $p_t \log p_s$. The notation $\langle \cdot, \cdot \rangle$ indicates an inner product, effectively applying the mask $\mathbf{M_t}$ to the element-wise KL divergence and aggregating the result.

**Phase 2: Classification head refinement.** The classification head is crucial for defining decision boundaries between new and existing tasks, as illustrated in Figure 2. Refining the classification head enhances learning outcomes while mitigating forgetting of previously acquired knowledge without disrupting the stable representations learned in the feature extractor.

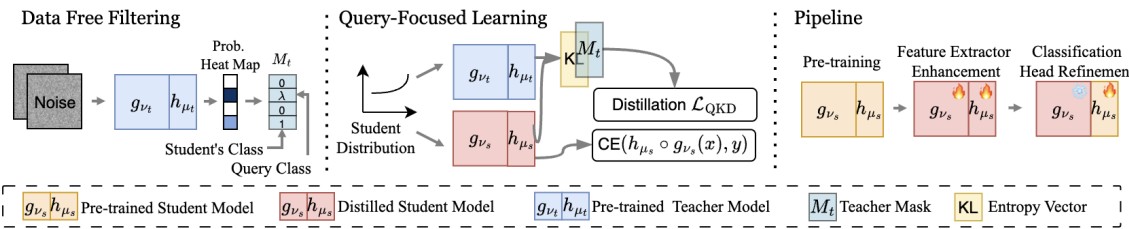

Figure 3: Query-based Knowledge Transfer (QKT): noise is applied to estimate the relevance of the teacher models to the student's query to obtain masks for each teacher model. Masked distillation is then applied to transfer the knowledge of the query classes from the teacher models to the student model. We then refine the classification head of the student model to prevent forgetting the previously learned knowledge.

To achieve this, we first perform *Head Replacement* by restoring the classification head $h_{\mu_s}$ to its state before Phase 1. This head is not influenced by the knowledge distilled from the teacher models, thereby preserving the classification patterns related to the knowledge that the student model has learned from its local data. Next, we freeze the feature extractor $g_{\nu_s}$ and only refine the classification head $h_{\mu_s}$ of the student model with the same strategy as in Equation 4. This dual approach ensures that the model effectively assimilates new knowledge from the query classes while retaining robust decision boundaries for previously learned tasks. Furthermore, the separation of training into two distinct phases provides enhanced stability, as observed in Appendix A.6.1. By isolating the refinements to the classification head, the risk of interference is reduced with stable feature representations. We shall show in Section 4.3 that this strategy significantly improves the model's performance on the query classes while maintaining the accuracy of the original tasks.

**Efficiency considerations.** QKT performs knowledge transfer in a single round, which is significantly more efficient than traditional centralized and peer-to-peer FL approaches that involve multiple communication rounds. The computation overhead for trainable parameters in Phase 2 is negligible; for instance, considering ResNet-18, its classification head comprises only about 0.04% of the total parameters. However, to reduce computational costs in Phase 1, especially in resource-constrained scenarios, we propose a simplified variant, termed **QKT Light**, that performs naive KD using all teachers and all classes to obtain a general feature extractor in Phase 1. Teacher filtering and class masking are only applied in Phase 2, where each client refines the classification head based on its specific needs.

By training a general feature extractor once and sharing the Phase 1 model among all clients, QKT Light substantially reduces the computational burden of Phase 1. Clients then proceed directly to Phase 2, where they only need to refine the classification head to adapt the general knowledge to their specific needs.

However, because QKT Light confines the query-focused knowledge transfer to the classification head, it may pose a higher risk of forgetting compared to the full QKT approach. Despite this, QKT Light still outperforms the existing solutions in meeting the diverse needs of query clients and is particularly effective for rapid adaptation scenarios.

## 4 EVALUATION

### 4.1 SETUP

**Task, datasets and models.** We evaluate our approach on image classification tasks using the following datasets: CIFAR10 (Krizhevsky, 2009), with 60,000 images across 10 classes; CIFAR100 (Krizhevsky, 2009), featuring 100 classes with 600 images per class, to test generalizability across more classes; CINIC10 (Darlow et al., 2018), which combines samples from ImageNet (Russakovsky et al., 2015) and CIFAR10, introducing natural distribution shifts (Luo et al., 2021); PathMNIST (Yang et al., 2023), a medical dataset containing 9 classes of colorectal cancer images; and BloodMNIST (Yang et al., 2023), featuring 8 classes of blood cell microscope images. Our experiments assume a distributed cross-silo training scenario with 10 clients. We use the ResNet-18 architecture (He et al., 2016) and the Adam optimizer (Kingma & Ba, 2017). More details are in Appendix A.2.

Table 2: Average accuracy on various datasets. The best are in bold, and the second-best are underlined.

| | | CIFAR10 | | CIFAR100 | | CINIC10 | | BloodMNIST | | PathMNIST | | Average | |
|---|---|---|---|---|---|---|---|---|---|---|---|---|---|
| | Comm | Path | Dir | Path | Dir | Path | Dir | Path | Dir | Path | Dir | Path | Dir |
| **Single Class Query** | | | | | | | | | | | | | |
| FedAvg | 100 | 52.33 | 60.55 | 53.12 | 37.59 | 22.30 | 30.39 | 43.34 | 63.13 | 49.88 | 60.81 | 44.19 | 50.49 |
| FedProx | 100 | 49.69 | 61.15 | 53.63 | 41.57 | 24.22 | 32.31 | 63.65 | 72.75 | 62.12 | 63.74 | 50.66 | 54.30 |
| Moon | 100 | 43.17 | 50.28 | 31.50 | 19.09 | 40.99 | 39.50 | 72.97 | 62.04 | 60.06 | 67.44 | 49.74 | 47.67 |
| FT-FedAvg | 100 | 46.20 | 49.15 | 34.95 | 34.19 | 45.55 | 44.18 | 46.99 | 60.98 | 44.71 | 55.28 | 43.68 | 48.76 |
| FedAvg(1) | 1 | 10.95 | 08.73 | 00.50 | 00.95 | 08.86 | 03.73 | 14.88 | 19.78 | 11.14 | 00.51 | 09.27 | 06.74 |
| Ensemble | 1 | 24.84 | 49.92 | 34.83 | 23.70 | 30.22 | 43.61 | 56.08 | 69.35 | 60.03 | 59.86 | 41.20 | 49.29 |
| PFNM | 1 | 10.66 | 03.29 | 00.54 | 00.37 | 19.27 | 23.83 | 08.85 | 20.15 | 22.21 | 11.90 | 12.31 | 11.91 |
| CLUE | 1 | 07.66 | 15.90 | 06.40 | 00.97 | 26.23 | 12.47 | 20.67 | 19.29 | 14.07 | 11.96 | 15.01 | 12.12 |
| KD | 1 | 51.97 | 51.00 | 39.84 | 29.39 | 49.64 | 50.64 | 43.77 | 56.40 | 58.18 | 32.40 | 48.68 | 43.97 |
| **QKT Light** | 1 | **75.78** | 61.44 | 68.17 | 49.37 | **71.27** | 70.41 | **78.15** | 74.70 | 77.25 | **78.31** | 74.12 | 66.85 |
| **QKT** | 1 | 74.56 | **71.35** | **68.48** | **51.42** | 71.02 | **73.50** | 77.65 | **75.23** | **83.41** | 77.95 | **75.02** | **69.89** |
| **Multi-Class Query** | | | | | | | | | | | | | |
| FedAvg | 100 | 51.27 | 59.92 | 46.29 | 43.90 | 27.95 | 46.17 | 53.33 | 73.76 | 41.35 | 57.51 | 44.04 | 56.25 |
| FedProx | 100 | 51.90 | 59.42 | 46.21 | 47.78 | 27.56 | 46.55 | 67.94 | **81.78** | 49.92 | 59.85 | 48.71 | 59.08 |
| Moon | 100 | 41.58 | 49.15 | 22.44 | 24.72 | 38.30 | 46.16 | **72.45** | 71.80 | 47.16 | 62.80 | 44.39 | 50.93 |
| FT-FedAvg | 100 | 28.35 | 33.23 | 12.50 | 25.09 | 27.80 | 45.20 | 32.29 | 45.28 | 25.87 | 38.09 | 25.36 | 37.38 |
| FedAvg(1) | 1 | 10.95 | 06.63 | 00.11 | 00.79 | 11.06 | 03.75 | 17.09 | 14.73 | 21.12 | 00.75 | 12.07 | 05.33 |
| Ensemble | 1 | 37.38 | 49.14 | 39.67 | 30.77 | 26.31 | 39.78 | 61.15 | 64.19 | 44.63 | 60.37 | 45.13 | 48.85 |
| PFNM | 1 | 08.46 | 06.89 | 01.70 | 01.13 | 19.09 | 12.75 | 17.43 | 15.49 | 16.43 | 11.51 | 12.62 | 09.55 |
| KD | 1 | 43.04 | 40.24 | 22.05 | 25.37 | 39.53 | 37.57 | 38.04 | 52.87 | 44.40 | 33.93 | 37.41 | 38.00 |
| **QKT Light** | 1 | 65.28 | 58.44 | 54.60 | 48.27 | 61.71 | 51.06 | 67.16 | 64.26 | 50.62 | 67.11 | 59.87 | 57.83 |
| **QKT** | 1 | **65.60** | **61.08** | **54.98** | **48.46** | **62.13** | **54.57** | 69.52 | 63.14 | **67.03** | **69.32** | **63.57** | **59.31** |

**Data distribution.** To simulate non-IID data distribution among clients, we adopt two commonly used schemes: **Pathological non-IID** (Path) (McMahan et al., 2017; Qin et al., 2023; Luo & Wu, 2022; Huang et al., 2021), where each client receives samples from $M$ exclusive classes with a random number of samples per class (with $M = 3$ by default), and **Dirichlet distributions** (Dir) (Yurochkin et al., 2019; Wang et al., 2020b), where the proportion $p_{i,s}$ of samples from class $i$ assigned to client $s$ is drawn from $\text{Dir}_C(\alpha)$ (with $\alpha = 0.1$ by default). We also vary $M$ and $\alpha$ to explore different heterogeneity levels in the ablation study (Table 5), with further details in Appendix A.3.

**Baselines.** We evaluate our QKT framework against several established baselines. This includes the widely used FL method, **FedAvg** (McMahan et al., 2017), and its one-round variant **FedAvg(1)**, FL methods like **FedProx** (Li et al., 2020b), and **Moon** (Li et al., 2021b), specifically designed to handle heterogeneous data distributions. We also incorporate **FT-FedAvg** (Wang et al., 2020a; Yu et al., 2022), a strong baseline for personalized FL methods (Jiang & Lin, 2023). We further compare QKT with methods that achieve knowledge transfer in a single peer-to-peer communication round. These include the **Ensemble** method (Dietterich, 2000), which combines predictions from multiple teacher models by averaging them; **PFNM** (Yurochkin et al., 2019), where local models are matched and merged into a global model using a probabilistic framework; **CLUE** (Daga et al., 2023), which dynamically integrates significant parameters from a helper model into the target model through multi-model boosting; and **KD** (Hinton et al., 2015), which performs naive distillation from all teacher models. We also assess a lighter variant of our approach, **QKT Light**, which simplifies the first phase of QKT by replacing Equation 4 with Equation 1 (the naive KD loss), and applies filtering and masking only in Phase 2 when refining the classification head. In Table 2, we evaluate the version where Phase 1 is performed locally by each client, without a central coordinator. Further details and comparisons of additional QKT Light variants are provided in Appendix A.4.

**Learning objectives.** In our experiments, each client issues a query to learn or improve a *single class* or *multiple classes*. We evaluate both scenarios for each client. The specific classes and, in the case of multi-class queries, the number of classes, are both selected randomly from the client's data distribution. Classes are chosen from the set of underrepresented classes in the client's data based on a predefined sample threshold (50 samples by default). The selection process follows a uniform distribution, ensuring that each eligible class has an equal probability of being selected.

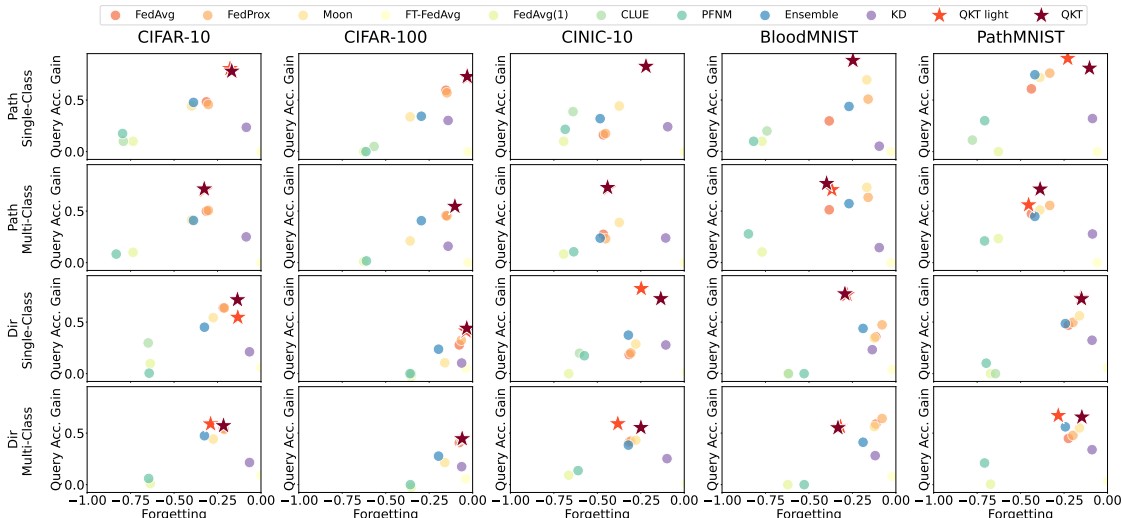

Figure 4: Forgetting vs. Query Accuracy Gain across datasets. Each column shows one dataset, while rows correspond to different data distributions (Pathological/Dirichlet) and query types (Single-Class/Multi-Class). Points closer to the upper-right corner represent lower forgetting and higher query accuracy gain.

**Evaluation metrics.** For any dataset $\mathcal{D}$, we define a subset $\mathcal{D}_j = \{(x, y) \in \mathcal{D}, y = j\}$ containing all the samples from class $j$; then the per-class accuracy of a model $f_\theta$ is $\mathrm{acc}(j) = \sum_{(x,y)\in\mathcal{D}_j} \mathbb{I}_{\mathrm{argmax}_k f_\theta(x)[k]=y}/|\mathcal{D}_j|$. For each client, we compute weighted average per-class accuracy as *Average Accuracy*, that is, $\mathrm{acc} = \frac{1}{\sum_{j\in\mathcal{Y}_S\cup\mathcal{Y}_Q} w_j} \sum_{j\in\mathcal{Y}_S\cup\mathcal{Y}_Q} w_j \mathrm{acc}(j)$, where $w_j$ is the weight of class $j$, $\mathcal{Y}_S$ is set of local classes and $\mathcal{Y}_Q$ is the set of query classes (Chen & Chao, 2021; Dai et al., 2023; Yu et al., 2022). The reported average accuracy (*Acc*) is the averaged $\mathrm{acc}(\cdot)$ across all clients' models. We also report the improvement in query class accuracy after knowledge transfer as *Query Acc. Gain*, and the decrease in accuracy on local classes post-learning as *Forgetting*. More details are in Appendix A.2. Finally, *Comm* represents the number of communication rounds required. For centralized FL methods, we follow a standard cross-silo FL setting where each round involves all clients sending their models to the server, followed by receiving the updated global model for the next round. In the other peer-to-peer approaches, a round entails each client receiving model weights from all other clients.

## 4.2 MAIN RESULTS

Table 2 provide a comprehensive comparison of the performance of QKT with FL baselines, as well as computationally efficient matching and ensemble-based methods across a variety of datasets.

**QKT across tasks and datasets.** For single-class queries, QKT consistently outperforms existing methods, achieving average improvements of 17.28% points on Pathological and 11.25% points on Dirichlet distribution. When querying for multiple classes, QKT demonstrates superior performance, with up to 21.18% points improvement on Pathological and up to 8.02% points on Dirichlet distribution. The larger margin of improvement on the Pathological distribution suggests that while QKT performs exceptionally well in handling pathological data, where the distribution shift is more pronounced, it remains highly competitive even in more balanced scenarios like the Dirichlet distribution. Figure 4 further visualizes the learning of query classes versus the forgetting of local classes for QKT and the baselines. The results show that QKT achieves a better balance between learning and forgetting compared to the existing methods.

**QKT vs. FL.** Existing FL methods, such as FedAvg, FedProx, and Moon, are designed to optimize a single global model that can generalize across all clients. They perform reasonably well in both single-class and multi-class query scenarios but fall short of QKT by an average of 15.32% points in single-class and 5.78% points in multi-class queries, highlighting QKT's advantage in focusing on queried classes. The only

Table 3: Ablation study of QKT components.

| Method | Acc | Query Acc. Gain | Forgetting |
|---|---|---|---|
| KD | 51.97 | 23.56 | -8.43 |
| KD + $\mathcal{T}_{\mathcal{Q}}$ | 63.13 | 81.55 | -38.21 |
| KD + $\mathcal{T}_{\mathcal{Q}}$ + $\mathbf{M_t}$ | 69.37 | 66.11 | -15.02 |
| QKT | 74.56 | 77.97 | -16.81 |

Table 4: Impact of $\lambda$.

| Lambda | Acc | Query Acc. Gain | Forgetting |
|---|---|---|---|
| $\lambda = 1$ | 69.77 | 58.52 | -9.71 |
| $\lambda = 1.5$ | 74.56 | 77.97 | -16.81 |
| $\lambda = 2$ | 74.31 | 87.11 | -24.62 |
| $\lambda = 4$ | 68.47 | 96.59 | -42.73 |

Table 5: Average accuracy in different levels of data heterogeneity.

| Method | Path(M=4) | Path(M=2) | Dir($\alpha$=0.01) |
|---|---|---|---|
| FedAvg | 53.96 | 12.29 | 29.57 |
| Ensemble | 49.12 | 16.04 | 27.29 |
| KD | 51.62 | 39.92 | 37.06 |
| QKT | **77.65** | **54.72** | **55.95** |

exception is BloodMNIST in the multi-class query scenario, where FL methods outperform QKT. This is likely due to the well-defined visual characteristics of the blood cell types, which allow the global model to generalize effectively even with limited training examples per class. However, when querying single classes in BloodMNIST, FL methods struggle with irrelevant classes, emphasizing the value of QKT's targeted learning. Additionally, FL methods require multiple communication rounds to converge, whereas QKT completes learning in a single round, making it more efficient in communication-limited environments. We also include FT-FedAvg, a strong personalized FL baseline (Jiang & Lin, 2023). While personalized methods are inherently limited in addressing client needs for under-represented classes, our results confirm these limitations, which similarly apply to other personalized or clustered FL approaches. For a fair comparison, we also include FedAvg(1), which performs only a single communication round. As expected, FedAvg(1) performs significantly worse due to insufficient communication, underscoring the limitations of FL in such settings.

**QKT vs. ensemble and matching-based methods.** In comparison, ensemble and matching-based methods, such as Ensemble, PFNM, and CLUE, are designed for efficient knowledge transfer in a single round. However, these methods perform significantly worse than QKT. The primary limitation of these methods is their dependence on model architecture and the homogeneity of data distributions. Additionally, they do not account for irrelevant knowledge, which is a key focus in QKT's design. Among single-round methods, KD performs reasonably well but suffers from forgetting issues, particularly in multi-class queries, as we discussed in Section 3.2. This demonstrates the importance of managing the learning-forgetting trade-off, which QKT addresses more effectively.

QKT Light generally outperforms existing methods, effectively addressing the diverse needs of query clients. Although there is a performance gap between QKT Light and the full QKT in some scenarios, QKT Light is well-suited for rapid adaptation in environments with limited computational resources.

## 4.3 ABLATION AND ANALYSES

We conducted ablation studies on the CIFAR10 dataset using a pathological distribution and a single-class query to evaluate various aspects of the QKT method.

**Ablation study of QKT components.** Table 3 shows the ablation of the QKT method. We compare the naive KD, the KD with only teacher set in Equation 2 (KD + $\mathcal{T}_{\mathcal{Q}}$), the KD + Mask in Equation 3 (KD + $\mathcal{T}_{\mathcal{Q}}$ + $\mathbf{M_t}$), and our QKT method with masked distillation and two-phase training. The results show that the full QKT method, which encompasses both feature extractor enhancement (Phase 1) and classification head refinement (Phase 2), outperforms these configurations in both accuracy and query accuracy gain, while effectively managing the trade-off with forgetting.

**Impact of $\lambda$.** Table 4 explores the effect of varying the $\lambda$ parameter, which determines the weight of the loss contribution for query classes within the mask $M_s$. This parameter balances the trade-off between learning new query-specific knowledge and retaining previously learned information. Our results demonstrate that a value of $\lambda$ between 1.5 and 2 provides a good balance between learning and forgetting across most datasets. For more complex datasets like CIFAR100, slightly increasing $\lambda$ (between 2.5 and 3.5) allows QKT to better prioritize query learning while still mitigating forgetting. Notably, even without tuning, the default $\lambda = 1$ remains effective, demonstrating QKT's robustness to this parameter under typical settings.

We propose an additional approach detailed in Appendix A.5, to control the balance between learning and forgetting. This approach, *Selective Weight Masking for Mitigating Forgetting*, uses weight importance

detection to selectively freeze critical weights in the classification head during Phase 2 of QKT. We included this in the appendix due to space limitations as mitigating forgetting is not the primary focus of this paper.

**Effect of data heterogeneity.** Table 5 presents the performance across varying levels of heterogeneity, highlighting QKT's consistent superiority over other methods. Main baselines are shown here, with additional baselines and detailed results provided in Appendix A.3.

**Effectiveness of noise-based filtering.** In Fig. 5, the left heatmap shows the normalized actual data distribution, making it easier to compare with the average predicted

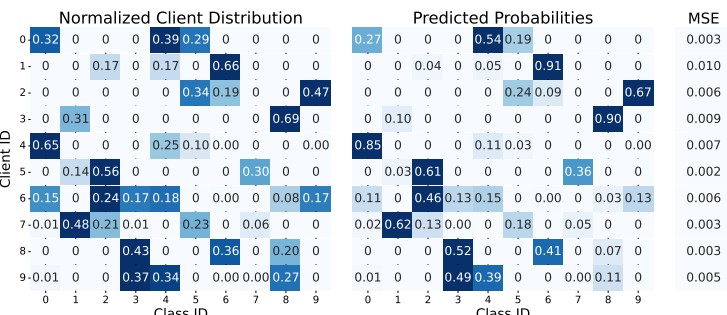

Figure 5: Comparison of actual data distribution (left) and predicted probabilities (right) for each client using noise-based filtering. The left heatmap is normalized to facilitate comparison, and the prediction error is shown in MSE.

probabilities (right) generated by inputting a batch of noise (20 samples per model). Each noise sample was shaped according to the model's input dimensions to ensure compatibility with the teacher models' architecture. We also report the Mean Squared Error (MSE) to measure prediction error. The low MSE values across all clients confirm the effectiveness of our noise-based method in identifying the classes each client model was trained on.

By setting an appropriate relevance threshold for these predicted probabilities, we can estimate which teacher models are most pertinent to a given class. Across various datasets, we found that a threshold of 0.01 effectively detected the meaningful presence of a class in a teacher's training data. This threshold can be adjusted to control the sensitivity: a higher threshold reduces false positives but may exclude some relevant teachers, while a lower threshold increases inclusiveness but may introduce noise. When the predicted probability for a class exceeds the pre-defined threshold, it indicates that the teacher has substantial training data for that class, making it a valuable source of knowledge for the student's learning process. This selective filtering ensures that the student model focuses on learning only from the most relevant parts of the teacher models' knowledge, enhancing learning efficiency and minimizing interference from irrelevant information.

Additional ablation studies, including QKT's scalability with more clients and the impact of different model architectures, are presented in Appendix A.6.

# 5    CONCLUSION

We focused on the problem of customized queries in decentralized collaborative learning with heterogeneous data, privacy concerns, and communication efficiency. To this end, we introduced a Query-based Knowledge Transfer (QKT) framework to enable query-specific knowledge distillation from teacher models, where a data-free masking strategy is employed to filter out irrelevant knowledge to prevent knowledge inference and staged training is applied to mitigate the forgetting in task-specific parameters. Our extensive experiments on both standard and clinical benchmarks demonstrated that QKT consistently outperforms state-of-the-art methods. By effectively addressing issues like knowledge interference and catastrophic forgetting, QKT offers a robust solution for decentralized learning while minimizing communication overhead. Future work may explore the application of QKT in real-time and highly dynamic scenarios, pushing the boundaries of what decentralized models can achieve in practical, privacy-preserving settings.

## ACKNOWLEDGMENTS

This publication is based upon work supported by the King Abdullah University of Science and Technology (KAUST) Office of Research Administration (ORA) under Award No. ORA-CRG2021-4699, and partly supported by UK Research and Innovation (UKRI) - Engineering and Physical Science Research Council (EPSRC) under grant No. EP/X035085/1. We are thankful to the anonymous reviewers for their thoughtful comments and suggestions that helped improving our paper. For computer time, this research used the resources of the Supercomputing Laboratory at KAUST.

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

# A APPENDIX

## A.1 EXTENDED DISCUSSION ON RELATED WORK

**Model Compression, Quantization, and Sparsification in FL.** To address communication costs in federated learning, techniques like model compression, quantization, and sparsification aim to reduce communication costs in FL by minimizing update sizes (Konečný et al., 2017; Sattler et al., 2019), but they can pose challenges in maintaining model accuracy and ensuring efficient convergence across heterogeneous clients.

**Replay-Based Continual Learning and Transfer Learning.** Data-free continual learning and transfer learning methods (Li & Hoiem, 2017; Kirkpatrick et al., 2017; Zenke et al., 2017) focus on preserving previously learned knowledge while adapting to new tasks. However, these methods assume access to a well-defined source and target, making them less suitable for decentralized learning environments with data privacy concerns and high heterogeneity.

**Peer-to-Peer Collaborative Learning Enhancements.** Cartel (Daga et al., 2019) enhances peer-to-peer collaborative learning by enabling dynamic task-specific interactions among nodes with similar workloads. This personalization improves model adaptability and efficiency under changing data and resource conditions. However, its focus on shallow models and metadata sharing limits its applicability for deeper, heterogeneous models typical in FL scenarios. CLUE (Daga et al., 2023), an extension of Cartel, introduces multi-modal boosting to dynamically integrate significant parameters from helper models into learner models. While effective in controlled environments, CLUE suffers from performance degradation when applied to models trained on divergent data distributions, exacerbating issues of forgetting and irrelevant parameter integration.

## A.2 EXPERIMENTAL DETAILS

**Baselines Implementation Details.**

- For all experiments, we use the Adam optimizer (Kingma & Ba, 2017) with a learning rate of $1 \times 10^{-3}$, a weight decay of $4 \times 10^{-4}$, and a batch size of 32, consistent with prior studies (Meng et al., 2023; Alballa & Canini, 2023).

- During local training, each client's model is pre-trained on its local dataset for up to 100 epochs, with early stopping applied if validation performance does not improve for 10 consecutive epochs.

- In FL approaches, the number of local training epochs per communication round ($E$) is set to 2.

- For generalized FL approaches (FedAvg, FedProx, Moon, FedAvg(1)), we adhere to the standard practice in personalized FL, where each client independently evaluates the global model, and the average accuracy across clients is reported.

- The hyperparameters used for each baseline generally follow the values recommended in their respective original papers: for FedProx, $\mu$ is set to 0.01; for Moon, $\mu$ is set to 5 and the temperature ($T$) is set to 0.5. In FT-FedAvg, $2 \times E$ local epochs are performed after executing FedAvg for 100 rounds, and the resulting average test accuracies are reported.

- For all KD-based approaches (naive KD and QKT), we use a default $\alpha$ parameter and temperature of 1, and use the student's data as the transfer set. Training is conducted over $E$ epochs using the transfer set, where $E$ is set to 25 for CIFAR10 and CINIC10, and 10 for all other datasets. To maintain simplicity, the same $E$ is used across all clients and both phases of full QKT. However, we explore varying $E$ in Appendix A.6.4 to highlight potential areas for further improvement. For QKT Light, $E$ is set to 5 for Phase 2.

**Evaluation Metrics.**

- *Average Accuracy:* For each client, we have the weighted average of per-class accuracy of query classes and local classes *i.e.*,

$$\text{acc} = \frac{\sum_{j \in \mathcal{Y}_S \cup \mathcal{Y}_Q} w_j \text{acc}(j)}{\sum_{j \in \mathcal{Y}_S \cup \mathcal{Y}_Q} w_j},$$

where $\mathcal{Y}_S$ is set of local classes, $\mathcal{Y}_Q$ is the set of query classes, $\text{acc}(j)$ for the query and local class $j$ is the per-class accuracy, and $w_j$ is the weight of class $j$. If the test set is balanced, for local classes $w_j$ is determined based on the class ratio in the client's training dataset, and is set to 1 for query classes (Chen & Chao, 2021; Dai et al., 2023; Yu et al., 2022). In other words, Average Accuracy for each client is the weighted sum of per-class accuracy, normalized by the summation of the weights. The overall average accuracy (Acc) is then computed by averaging *acc* across all clients' models.

- *Query Acc. Gain:* It measures the improvement in query class accuracy after knowledge transfer. For a query set $\mathcal{Q}$, Query Acc. Gain is computed as

$$\frac{1}{|\mathcal{Q}|} \sum_{i \in \mathcal{Q}} (\text{acc}_{\text{post}}(i) - \text{acc}_{\text{pre}}(i)),$$

where "post" and "pre" denote post- and pre-learning per-class accuracy, respectively.

- *Forgetting:* It measures the decrease in accuracy on local classes post-learning relative to pre-learning. For the student's label set $\mathcal{Y}_S$, the forgetting is measured by

$$\frac{\sum_{j \in \mathcal{Y}_S} \min(0, \text{acc}_{\text{post}}(j) - \text{acc}_{\text{pre}}(j))}{|\mathcal{Y}_S|}$$

- *Uniform Accuracy:* This additional evaluation metric follows the approach suggested by (Dai et al., 2023), setting the weight $w_j$ to 1 for all classes while using a uniform test set. This adjustment ensures that each class is treated with equal importance, distinct from the original accuracy metric (Acc), which prioritizes local classes based on their prevalence in the local data distribution. While the uniform accuracy metric offers a measure of a model's generalization ability across all classes, its utility may be limited in highly imbalanced data distributions. This limitation is due to our specific objective, which deviates from generalized methods that aim for universal class representation. Our primary objective with QKT is to enhance knowledge about query classes while minimizing the forgetting of existing knowledge, which naturally reflects the data distribution of local classes. For example, in the Dir distribution, certain classes may contain only a single data point, making it unrealistic to expect models to gain significant knowledge about these classes if they are not represented enough in the local data and are not part of the query.

  It is worth highlighting that QKT still delivers strong uniform accuracy performance under the Path distribution, showcasing its robustness. However, its performance in the Dir distribution is understandably lower, aligning with our expectations and our specific objective.

## A.3 QKT PERFORMANCE ACROSS DIFFERENT LEVELS OF DATA HETEROGENEITY

This section provides the results of our experiments across different levels of data heterogeneity, using both Pathological and Dirichlet distribution schemes.

| Method | Acc | Query Acc. Gain | Forgetting | Uniform Acc. |
|---|---|---|---|---|
| **Single-Class Queries** | | | | |
| FedAvg | 53.96 | 54.49 | -27.24 | 58.62 |
| FedProx | 58.62 | 60.52 | -24.22 | 62.04 |
| Moon | 46.51 | 44.00 | -30.78 | 52.06 |
| CLUE | 12.30 | 10.00 | -72.15 | 11.15 |
| Ensemble | 49.12 | 48.36 | -31.99 | 52.47 |
| KD | 51.62 | 31.50 | -12.12 | 63.80 |
| QKT | **77.65** | 87.59 | -17.25 | 69.86 |
| **Multi-Class Queries** | | | | |
| FedAvg | 51.97 | 54.29 | -27.24 | 55.25 |
| FedProx | 56.09 | 58.37 | -24.22 | 58.86 |
| Moon | 47.83 | 50.70 | -30.78 | 50.19 |
| Ensemble | 57.48 | 61.96 | -31.99 | 55.58 |
| KD | 44.98 | 34.49 | -12.12 | 56.13 |
| QKT | **64.84** | 72.70 | -31.99 | 58.17 |

Table 6: Detailed results for the Pathological distribution with $M = 4$.

| Method | Acc | Query Acc. Gain | Forgetting | Uniform Acc. |
|---|---|---|---|---|
| **Single-Class Queries** | | | | |
| FedAvg | 12.29 | 4.91 | -68.40 | 16.21 |
| FedProx | 16.62 | 12.72 | -67.00 | 19.87 |
| Moon | 39.41 | 26.72 | -52.36 | 33.54 |
| CLUE | 9.13 | 17.33 | -85.38 | 9.11 |
| Ensemble | 16.04 | 14.58 | -73.71 | 15.18 |
| KD | 39.92 | 13.50 | -22.55 | 50.16 |
| QKT | **54.72** | 44.00 | -26.20 | 58.36 |
| **Multi-Class Queries** | | | | |
| FedAvg | 12.63 | 9.64 | -68.40 | 14.57 |
| FedProx | 14.19 | 11.12 | -67.00 | 16.40 |
| Moon | 29.78 | 27.03 | -52.36 | 31.75 |
| Ensemble | 25.12 | 26.68 | -73.71 | 23.36 |
| KD | 22.89 | 12.38 | -22.55 | 30.49 |
| QKT | **33.72** | 33.70 | -53.63 | 34.11 |

Table 7: Detailed results for the Pathological distribution with $M = 2$.

| Method | Acc | Query Acc. Gain | Forgetting | Uniform Acc. |
|---|---|---|---|---|
| **Single-Class Queries** | | | | |
| FedAvg | 29.57 | 43.82 | -76.55 | 28.32 |
| FedProx | 44.80 | 58.38 | -59.06 | 39.99 |
| Moon | 42.46 | 59.56 | -65.79 | 38.52 |
| CLUE | 6.62 | 2.59 | -86.29 | 10.47 |
| Ensemble | 27.29 | 39.71 | -77.07 | 27.23 |
| KD | 37.06 | 9.77 | -29.65 | 31.38 |
| QKT | **55.95** | 27.80 | -11.44 | 44.19 |
| **Multi-Class Queries** | | | | |
| FedAvg | 17.39 | 18.44 | -76.55 | 17.67 |
| FedProx | 32.53 | 35.73 | -59.06 | 32.77 |
| Moon | 29.49 | 32.15 | -65.79 | 29.51 |
| Ensemble | 20.81 | 24.09 | -77.07 | 21.03 |
| KD | 22.79 | 6.13 | -29.65 | 22.14 |
| QKT | **34.75** | 23.89 | -41.55 | 29.92 |

Table 8: Detailed results for the Dirichlet distribution with $\alpha = 0.01$.

### A.4 QKT LIGHT VARIANTS

In this section, we explore the variants of **QKT Light**, a simplified version of our QKT framework, which replaces Query-Focused Learning with naive KD in Phase 1, and applies teacher filtering and class masking only during Phase 2 to refine the classification head for query-specific knowledge transfer.

Phase 1 of QKT Light can be implemented in several ways: (1) each client independently performs naive KD using local data, (2) a central server consolidates all models to perform naive KD once for all clients using an unlabeled dataset, or (3) a volunteer client with sufficient resources performs KD on behalf of others.

In Table 2 of the main text, we evaluate the version where Phase 1 is performed locally by each client, without a central coordinator. Although this variant does not reduce computational costs in the decentralized setup, it demonstrates the feasibility of building a general feature extractor before refining the classification head in Phase 2.

Detailed results of all three QKT Light variants, along with the QKT approach, are presented in the tables below. Notably, all QKT Light variants outperform traditional baselines across datasets, though they exhibit slightly increased forgetting in some setups compared to full QKT due to the simplified Phase 1 design.

| | | Acc | Query Acc. gain | Forgetting | Uniform Acc. |
|---|---|---|---|---|---|
| **Path** | **Single class queries** | | | | |
| | QKT | 74.56 | 77.97 | -16.81 | 74.28 |
| | QKT light Student data | 75.78 | 85.50 | -20.83 | 72.14 |
| | QKT light Centralized server | 75.75 | 88.15 | -23.51 | 70.63 |
| | QKT light Volunteer client | 74.46 | 86.01 | -24.00 | 69.61 |
| | **Multi-class queries** | | | | |
| | QKT | 65.60 | 71.40 | -32.63 | 60.13 |
| | QKT light Student data | 65.29 | 70.85 | -32.24 | 59.99 |
| | QKT light Centralized server | 66.83 | 73.84 | -36.37 | 59.98 |
| | QKT light Volunteer client | 66.83 | 73.84 | -36.37 | 59.98 |
| **Dir** | **Single class queries** | | | | |
| | QKT | 71.35 | 71.63 | -13.52 | 52.29 |
| | QKT light Student data | 61.44 | 54.61 | -13.32 | 47.80 |
| | QKT light Centralized server | 75.35 | 84.86 | -15.86 | 51.41 |
| | QKT light Volunteer client | 77.91 | 88.91 | -15.80 | 54.06 |
| | **Multi-class queries** | | | | |
| | QKT | 61.08 | 57.16 | -21.55 | 48.69 |
| | QKT light Student data | 58.44 | 58.87 | -28.98 | 46.27 |
| | QKT light Centralized server | 60.68 | 60.49 | -28.24 | 47.59 |
| | QKT light Volunteer client | 63.31 | 64.14 | -26.13 | 49.40 |

Table 9: Performance of QKT Light variants and QKT on *CIFAR10* for single-class and multi-class queries under both Pathological and Dirichlet distributions.

| | | Acc | Query Acc. gain | Forgetting | Uniform Acc. |
|---|---|---|---|---|---|
| **Path** | **Single class queries** | | | | |
| | QKT | 68.48 | 72.80 | -3.10 | 64.07 |
| | QKT light Student data | 68.17 | 71.60 | -3.02 | 64.41 |
| | QKT light Centralized server | 62.63 | 64.50 | -5.65 | 60.48 |
| | QKT light Volunteer client | 63.29 | 66.10 | -6.04 | 60.11 |
| | **Multi-class queries** | | | | |
| | QKT | 54.98 | 54.56 | -10.19 | 55.06 |
| | QKT light Student data | 54.60 | 54.16 | -10.37 | 54.87 |
| | QKT light Centralized server | 49.60 | 47.89 | -9.42 | 54.23 |
| | QKT light Volunteer client | 48.93 | 47.16 | -9.36 | 53.91 |
| **Dir** | **Single class queries** | | | | |
| | QKT | 51.42 | 44.00 | -3.36 | 33.39 |
| | QKT light Student data | 49.37 | 41.30 | -3.83 | 30.99 |
| | QKT light Centralized server | 53.62 | 53.20 | -5.56 | 32.43 |
| | QKT light Volunteer client | 54.91 | 55.10 | -5.28 | 32.65 |
| | **Multi-class queries** | | | | |
| | QKT | 48.46 | 44.50 | -6.01 | 36.62 |
| | QKT light Student data | 48.27 | 44.33 | -5.78 | 37.34 |
| | QKT light Centralized server | 47.18 | 43.78 | 7.45 | 36.29 |
| | QKT light Volunteer client | 48.64 | 45.34 | -6.82 | 37.25 |

Table 10: Performance of QKT Light variants and QKT on *CIFAR100* for single-class and multi-class queries under both Pathological and Dirichlet distributions.

| | | Acc | Query Acc. gain | Forgetting | Uniform Acc. |
|---|---|---|---|---|---|
| **Path** | **Single class queries** | | | | |
| | QKT | 71.02 | 82.68 | -22.02 | 65.42 |
| | QKT light Student data | 71.27 | 82.70 | -21.67 | 65.71 |
| | QKT light Centralized server | 68.02 | 83.00 | -26.33 | 61.95 |
| | QKT light Volunteer client | 70.17 | 85.37 | -24.76 | 64.28 |
| | **Multi-class queries** | | | | |
| | QKT | 62.13 | 72.80 | -44.10 | 49.80 |
| | QKT light Student data | 61.71 | 71.93 | -43.62 | 49.72 |
| | QKT light Centralized server | 60.67 | 70.82 | -44.20 | 48.58 |
| | QKT light Volunteer client | 62.17 | 72.46 | -43.47 | 50.22 |
| **Dir** | **Single class queries** | | | | |
| | QKT | 73.48 | 72.68 | -13.49 | 46.73 |
| | QKT light Student data | 70.14 | 82.50 | -24.77 | 42.62 |
| | QKT light Centralized server | 69.62 | 81.44 | -24.31 | 42.23 |
| | QKT light Volunteer client | 67.48 | 82.47 | -29.85 | 38.51 |
| | **Multi-class queries** | | | | |
| | QKT | 54.57 | 55.38 | -24.93 | 43.58 |
| | QKT light Student data | 51.01 | 59.07 | -38.21 | 38.72 |
| | QKT light Centralized server | 46.50 | 53.78 | -41.76 | 34.78 |
| | QKT light Volunteer client | 51.09 | 58.41 | -38.52 | 39.22 |

Table 11: Performance of QKT Light variants and QKT on *CINIC10* for single-class and multi-class queries under both Pathological and Dirichlet distributions.

|  |  | Acc | Query Acc. gain | Forgetting | Uniform Acc. |
|---|---|---|---|---|---|
| **Path** | **Single class queries** | | | | |
|  | QKT | 77.65 | 88.42 | -24.71 | 71.91 |
|  | QKT light Student data | 78.14 | 88.03 | -24.35 | 72.55 |
|  | QKT light Centralized server | 79.29 | 92.30 | -25.68 | 72.41 |
|  | QKT light Volunteer client | 79.39 | 90.19 | -24.03 | 72.97 |
|  | **Multi-class queries** | | | | |
|  | QKT | 69.52 | 76.72 | -39.85 | 62.67 |
|  | QKT light Student data | 67.16 | 70.73 | -36.81 | 63.11 |
|  | QKT light Centralized server | 69.83 | 77.89 | -40.25 | 62.62 |
|  | QKT light Volunteer client | 68.38 | 77.69 | -42.16 | 60.30 |
| **Dir** | **Single class queries** | | | | |
|  | QKT | 75.23 | 77.37 | -29.31 | 52.18 |
|  | QKT light Student data | 74.70 | 76.45 | -28.11 | 52.14 |
|  | QKT light Centralized server | 75.33 | 76.12 | -28.76 | 52.32 |
|  | QKT light Volunteer client | 76.60 | 76.24 | -26.35 | 53.45 |
|  | **Multi-class queries** | | | | |
|  | QKT | 63.12 | 55.12 | -33.31 | 54.16 |
|  | QKT light Student data | 64.26 | 56.16 | -31.87 | 55.53 |
|  | QKT light Centralized server | 66.67 | 59.45 | -33.05 | 56.96 |
|  | QKT light Volunteer client | 62.21 | 54.29 | -33.91 | 53.92 |

Table 12: Performance of QKT Light variants and QKT on *BloodMNIST* for single-class and multi-class queries under both Pathological and Dirichlet distributions.

|  |  | Acc | Query Acc. gain | Forgetting | Uniform Acc. |
|---|---|---|---|---|---|
| **Path** | **Single class queries** | | | | |
|  | QKT | 83.41 | 80.88 | -10.36 | 80.85 |
|  | QKT light Student data | 77.25 | 90.37 | -22.89 | 71.10 |
|  | QKT light Centralized server | 76.91 | 97.36 | -30.63 | 66.64 |
|  | QKT light Volunteer client | 79.25 | 95.16 | -27.57 | 69.63 |
|  | **Multi-class queries** | | | | |
|  | QKT | 67.03 | 71.28 | -38.76 | 62.88 |
|  | QKT light Student data | 50.62 | 55.97 | -45.38 | 47.38 |
|  | QKT light Centralized server | 47.58 | 55.11 | -51.33 | 43.83 |
|  | QKT light Volunteer client | 50.20 | 54.44 | -45.33 | 46.99 |
| **Dir** | **Single class queries** | | | | |
|  | QKT | 77.95 | 72.56 | -14.95 | 54.54 |
|  | QKT light Student data | 78.13 | 73.71 | -15.36 | 55.29 |
|  | QKT light Centralized server | 78.26 | 74.16 | -15.77 | 55.21 |
|  | QKT light Volunteer client | 77.10 | 69.60 | -13.86 | 55.87 |
|  | **Multi-class queries** | | | | |
|  | QKT | 69.32 | 65.52 | -14.75 | 54.82 |
|  | QKT light Student data | 67.11 | 67.01 | -28.33 | 50.40 |
|  | QKT light Centralized server | 65.27 | 64.86 | -24.61 | 49.28 |
|  | QKT light Volunteer client | 66.56 | 66.12 | -24.62 | 51.38 |

Table 13: Performance of QKT Light variants and QKT on *PathMNIST* for single-class and multi-class queries under both Pathological and Dirichlet distributions.

A.5  SELECTIVE WEIGHT MASKING FOR MITIGATING FORGETTING.

To help mitigate forgetting during the QKT we explore a novel approach that can be applied during Phase 2 in QKT. The approach utilizes *Weight Importance Detection* to identify the important weights in the classification head $h_{\mu_s}$ based on their contribution to the student's original tasks. This is achieved by calculating L2 norm-based importance scores for each weight, derived from gradients during backpropagation and averaged over the student's training batches. Once the important weights are identified, we apply *Weight Masking*. Masks are created based on the top Z% of the importance scores. During this process, the important weights are frozen to mitigate forgetting, while the gradients of less important weights are preserved. This selective adjustment allows the model to focus on learning the query task without compromising the integrity of the original task. Finally, *Fine-tuning* is performed with the feature extractor $g_{\nu_s}$ frozen and only the classification head $h_{\mu_s}$ being refined, ensuring the model effectively incorporates new knowledge from the query task while maintaining the stability of the learned feature representations from Stage 1.

It is important to note that this strategy is not aimed at enhancing overall performance but rather at managing the balance between learning and forgetting, thereby achieving similar performance while mitigating forgetting.

Figure 6 illustrates the trade-off between learning and forgetting for different values of Z%, showing how different levels of weight masking affect both query accuracy gain and forgetting. The results, summarized in Table 14, demonstrate that increasing the masking percentage (Z%) effectively reduces forgetting but can lead to a decrease in query accuracy gain. This highlights the inherent balance that must be managed between learning new tasks and retaining previously learned information.

| Z% | Acc. | Query Acc. gain | Forgetting |
|----|------|-----------------|------------|
| 0 | 74.26 | 81.74 | -20.01 |
| 0.5 | 73.93 | 77.90 | -17.43 |
| 1 | 73.03 | 71.79 | -14.23 |
| 2.5 | 66.67 | 52.63 | -9.49 |
| 5 | 54.73 | 26.67 | -04.67 |
| 10 | 46.63 | 03.17 | -01.60 |

Table 14: Effect of different masking percentages (Z%) on the trade-off between query accuracy gain, and forgetting during QKT.

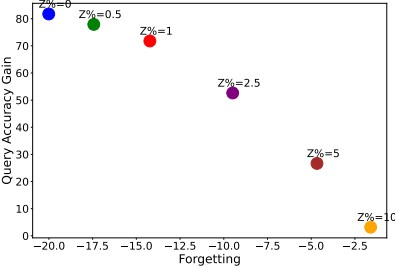

Figure 6: Trade-off between query accuracy gain vs. forgetting for different values of Z%. Each point represents a specific masking percentage, illustrating how different levels of weight masking affect both query accuracy gain and forgetting.

## A.6 ADDITIONAL ABLATION STUDIES

### A.6.1 DYNAMIC PERFORMANCE DURING QKT PHASES

This subsection analyzes the dynamic performance of the two phases in QKT. While Table 2 demonstrates the overall performance of the two stages, we further compare the validation accuracy after each epoch during Phase 1 and Phase 2 for a random subset of clients trained on CIFAR10. The plot in Figure 7 also includes the initial local accuracy before Phase 1 for reference.

- **Performance Improvement Across Phases:** Both Phase 1 and Phase 2 of QKT show substantial improvements over the local accuracy baseline (orange). These results highlight the effectiveness of QKT in leveraging knowledge transfer to enhance model performance.

- **Phase 2 Stability:** Phase 2 exhibits remarkable stability compared to Phase 1, as freezing the feature extractor preserves its learned representations. By focusing exclusively on classification head refinement, Phase 2 avoids the fluctuations observed in Phase 1 caused by simultaneous updates to both the feature extractor and classification head.

This analysis highlights how the two-phase design balances adaptability and stability, addressing the challenges of heterogeneous collaborative learning environments.

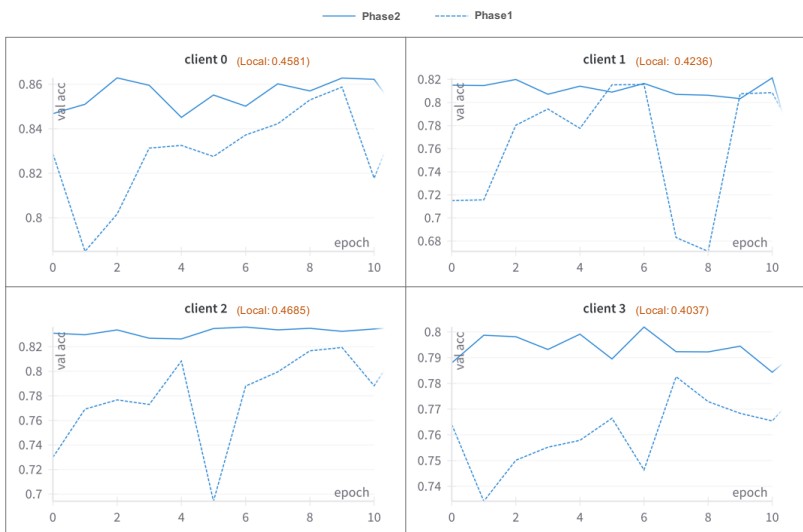

Figure 7: Dynamic validation accuracy across epochs during Phase 1 and Phase 2 for a subset of clients trained on CIFAR10. Initial local accuracy (orange) is also shown for reference.

### A.6.2 PERFORMANCE EVALUATION WITH INCREASING NUMBER OF CLIENTS

To evaluate QKT's performance with a larger number of clients, we expanded our experiments to include 50 clients using CIFAR10 under a pathological distribution, building on the experimental setup described in Section 4.1. The same evaluation metrics outlined in Appendix A.2 were used to ensure consistency with our main experiments.

We selected the main baselines that achieved competitive results in earlier experiments. To handle the increased number of clients, we adopted typical participation rates ($P\%$) of 10% (Table 15) and 20% (Table 16), as commonly used in cross-device federated learning. For FL methods, this represents the number of clients selected for each round, while for single-round methods like QKT, it represents the number of clients participating in the single knowledge transfer round. Additionally, the "Ensemble (all models)" baseline was included to assess performance when utilizing all client models simultaneously.

At a larger scale, with 50 clients, QKT consistently outperformed other baselines, achieving higher accuracy, greater query accuracy gain, and lower forgetting rates. Moreover, QKT's single-round knowledge transfer significantly reduces communication, storage, and computational demands compared to traditional FL methods.

| Participation Rate (P%) | | 10% | | |
|---|---|---|---|---|
| **Method** | **Acc.** | **Query Acc. Gain** | **Forgetting** | **Uniform Acc.** |
| **Single-Class Queries** | | | | |
| FedAvg | 0.355533 | 0.346319 | -0.421399 | 0.3601400 |
| Ensemble (all models) | 0.377580 | 0.391380 | -0.420330 | 0.3706799 |
| Ensemble | 0.224559 | 0.238380 | -0.580839 | 0.217650 |
| KD | 0.321850 | 0.158440 | -0.291293 | 0.40355 |
| QKT | **0.590206** | 0.512379 | -0.122973 | 0.629119 |
| **Multi-Class Queries** | | | | |
| FedAvg | 0.372590 | 0.377289 | -0.421399 | 0.3693679 |
| Ensemble (all models) | 0.361797 | 0.348241 | -0.420330 | 0.366202 |
| Ensemble | 0.2205406 | 0.224228 | -0.558339 | 0.217794 |
| KD | 0.270026 | 0.180399 | -0.291293 | 0.340036 |
| QKT | **0.442319** | 0.480412 | -0.318020 | 0.430587 |

Table 15: Performance comparison with 50 clients and participation rate ($P\%$) = 10%. Results are reported as average values for accuracy metrics.

| Participation Rate (P%) | | | 20% | |
| --- | --- | --- | --- | --- |
| **Method** | **Acc.** | **Query Acc. Gain** | **Forgetting** | **Uniform Acc.** |
| **Single-Class Queries** | | | | |
| FedAvg | 0.325116 | 0.304820 | -0.439373 | 0.335264 |
| Ensemble (all models) | 0.377580 | 0.391380 | -0.420330 | 0.3706799 |
| Ensemble | 0.28953 | 0.31402 | -0.524606 | 0.277285 |
| KD | 0.33096 | 0.235680 | -0.346913 | 0.378609 |
| QKT | **0.645558** | 0.652679 | -0.144859 | 0.642039 |
| **Multi-Class Queries** | | | | |
| FedAvg | 0.379309 | 0.399826 | -0.439373 | 0.365998 |
| Ensemble (all models) | 0.361797 | 0.348241 | -0.420330 | 0.366202 |
| Ensemble | 0.255061 | 0.252811 | -0.548939 | 0.254010 |
| KD | 0.28853 | 0.252986 | -0.395126 | 0.316938 |
| QKT | **0.442734** | 0.488701 | -0.410544 | 0.409805 |

Table 16: Performance comparison with 50 clients and participation rate ($P\%$) = 20%. Results are reported as average values for accuracy metrics.

### A.6.3 ABLATION STUDY ON MODEL ARCHITECTURES

In addition to our primary experiments using ResNet-18, we further investigate the impact of model architecture by evaluating two alternative models: a smaller model consisting of two convolutional layers followed by a fully connected layer, and a larger model using the ResNet-50 architecture. This ablation study enables us to analyze how model capacity influences the performance and stability of different methods. The results, shown in Tables 17 and 18, include the main baselines that achieved competitive results in the primary experiments.

The results illustrate the impact of model architecture on the performance and stability of different methods under Single-Class and Multi-Class query scenarios. With the smaller model, QKT achieves the highest accuracy and query accuracy gain across both query types, while maintaining a relatively low forgetting rate. This highlights QKT's effectiveness even with limited model capacity, outperforming the other baselines in both performance and stability. Moreover, with the larger model, QKT maintains its performance, showcasing the highest accuracy and query accuracy gains with the least variability.

In contrast, other methods, such as FedAvg and Ensemble, show varying levels of performance and stability depending on the model architecture, with Ensemble achieving moderate gains in accuracy but exhibiting instability in forgetting, especially with the larger model. These findings suggest that while model architecture plays a role, QKT's design offers inherent stability that is less dependent on model capacity than other methods, making it a versatile option for collaborative learning across a range of architectures.

| Method | Acc. | Query Acc. Gain | Forgetting | Uniform Acc. |
|---|---|---|---|---|
| **Single-Class Queries** | | | | |
| FedAvg | 0.5140 | 0.4715 | -0.2710 | 0.5727 |
| Ensemble | 0.4241 | 0.4293 | -0.4386 | 0.4420 |
| KD | 0.5482 | 0.3188 | -0.1207 | 0.6546 |
| QKT | **0.6905** | 0.7110 | -0.1869 | 0.7077 |
| **Multi-Class Queries** | | | | |
| FedAvg | 0.5530 | 0.5554 | -0.2710 | 0.5796 |
| Ensemble | 0.3702 | 0.3612 | -0.4386 | 0.3996 |
| KD | 0.4456 | 0.2855 | -0.1207 | 0.5538 |
| QKT | **0.5888** | 0.5702 | -0.2590 | 0.5868 |

Table 17: Performance comparison using a smaller model architecture. Results are reported as average values for accuracy metrics.

| Method | Acc. | Query Acc. Gain | Forgetting | Uniform Acc. |
|---|---|---|---|---|
| **Single-Class Queries** | | | | |
| FedAvg | 0.3256 | 0.3325 | -0.4478 | 0.3636 |
| Ensemble | 0.3868 | 0.3718 | 0.3836 | -0.4193 |
| KD | 0.4372 | 0.1943 | -0.1327 | 0.5662 |
| QKT | **0.6945** | 0.8010 | -0.2195 | 0.6554 |
| **Multi-Class Queries** | | | | |
| FedAvg | 0.3055 | 0.2716 | -0.4478 | 0.3429 |
| Ensemble | 0.3306 | 0.3255 | -0.4193 | 0.3531 |
| KD | 0.3890 | 0.2534 | -0.1310 | 0.4944 |
| QKT | **0.5573** | 0.5947 | -0.2893 | 0.5237 |

Table 18: Performance comparison for the larger model architecture (ResNet-50). Results are reported as average values for accuracy metrics.

### A.6.4 The effect of variable training epochs for each client

We explore using a variable $E$ for each client, tuned with a validation set consisting of 1% of the training data for CINIC10 and 10% for other datasets. Note that such validation sets are not assumed available in our main experiments; we present this exploration to highlight potential areas for further improvement.

| | | CIFAR10 | | CIFAR100 | | CINIC10 | | BloodMNIST | | PathMNIST | |
|---|---|---|---|---|---|---|---|---|---|---|---|
| | | Uniform E | Var E | Uniform E | Var E | Uniform E | Var E | Uniform E | Var E | Uniform E | Var E |
| Path | Single-class Q | 0.7456 | 0.7629 (↑ 2.3%) | 0.6848 | 0.6905 (↑ 0.8%) | 0.7102 | 0.7269 (↑ 2.3%) | 0.7765 | 0.7979 (↑ 2.8%) | 0.8341 | 0.8457 (↑ 1.4%) |
| | Multi-class Q | 0.6560 | 0.6746 (↑ 1.8%) | 0.5498 | 0.5631 (↑ 1.3%) | 0.6071 | 0.6283 (↑ 2.1%) | 0.6952 | 0.7148 (↑ 1.9%) | 0.6703 | 0.6814 (↑ 1.1%) |
| Dir | Single-class Q | 0.7135 | 0.7360 (↑ 2.3%) | 0.5142 | 0.5373 (↑ 2.3%) | 0.7348 | 0.7503 (↑ 1.6%) | 0.7523 | 0.8003 (↑ 4.8%) | 0.7795 | 0.7866 (↑ 0.9%) |
| | Multi-class Q | 0.6108 | 0.6272 (↑ 1.7%) | 0.4846 | 0.5037 (↑ 1.9%) | 0.5457 | 0.5610 (↑ 1.5%) | 0.6312 | 0.6686 (↑ 3.7%) | 0.6932 | 0.6987 (↑ 0.6%) |

Table 19: Improvement of Var E over Uniform E across various datasets, highlighting the gains observed in both single-class and multi-class queries for different distributions (Path and Dir).

## A.7 DETAILED RESULTS

| | | Acc | Query Acc. gain | Forgetting | Uniform Acc. |
|---|---|---|---|---|---|
| **Path** | **Single class queries** | | | | |
| | (Local: Acc = 44.87, Query class acc = 0.0) | | | | |
| | FedAvg | 52.33 | 48.26 | -31.40 | 50.29 |
| | FedProx | 49.69 | 45.63 | -30.28 | 54.70 |
| | Moon | 43.17 | 44.25 | -39.92 | 46.44 |
| | FT-FedAvg | 46.20 | 0.00 | 0.00 | 67.95 |
| | fedAvg(1) | 10.95 | 10.00 | -73.61 | 15.00 |
| | CLUE | 7.66 | 10.00 | -79.16 | 9.99 |
| | PFNM | 14.62 | 17.55 | -79.61 | 12.42 |
| | Ensemble | 46.59 | 47.71 | -38.83 | 48.31 |
| | KD | 51.97 | 23.56 | -8.43 | 65.91 |
| | QKT | 74.56 | 77.97 | -16.81 | 74.28 |
| | QKT light | 75.78 | 80.00 | -17.10 | 72.14 |
| | **Multi-class queries** | | | | |
| | (Local: Acc = 27.58, Query class acc = 0.0) | | | | |
| | FedAvg | 51.27 | 49.96 | -31.40 | 54.02 |
| | FedProx | 51.90 | 50.65 | -30.28 | 54.76 |
| | Moon | 41.58 | 41.44 | -39.92 | 44.35 |
| | FT-FedAvg | 28.36 | 0.00 | 0.00 | 48.54 |
| | fedAvg(1) | 10.95 | 10.00 | -73.61 | 15.00 |
| | PFNM | 6.79 | 8.30 | -83.27 | 5.83 |
| | Ensemble | 41.43 | 40.90 | -38.83 | 44.22 |
| | KD | 43.04 | 24.99 | -8.50 | 55.59 |
| | QKT | 65.60 | 71.40 | -32.63 | 60.13 |
| | QKT light | 65.29 | 70.85 | -32.24 | 59.99 |
| **Dir** | **Single class queries** | | | | |
| | (Local: Acc = 44.40, Query class acc = 0.21) | | | | |
| | FedAvg | 60.55 | 63.80 | -21.71 | 65.04 |
| | FedProx | 61.15 | 63.72 | -21.05 | 64.99 |
| | Moon | 50.28 | 54.03 | -27.48 | 55.09 |
| | FT-FedAvg | 49.16 | 5.66 | 0.00 | 55.95 |
| | FedAvg(1) | 8.73 | 9.79 | -63.71 | 7.88 |
| | CLUE | 15.90 | 29.79 | -64.87 | 11.77 |
| | PFNM | 3.29 | 0.48 | -64.32 | 7.29 |
| | Ensemble | 49.45 | 45.03 | -32.52 | 40.89 |
| | KD | 51.00 | 21.19 | -6.60 | 54.47 |
| | QKT | 71.35 | 71.63 | -13.52 | 52.29 |
| | QKT light | 61.44 | 54.61 | -13.32 | 47.80 |
| | **Multi-class queries** | | | | |
| | (Local: Acc = 25.98, Query class acc = 3.04) | | | | |
| | FedAvg | 59.92 | 54.78 | -21.71 | 62.10 |
| | FedProx | 59.51 | 53.62 | -21.05 | 62.04 |
| | Moon | 49.46 | 44.29 | -27.48 | 52.13 |
| | FT-FedAvg | 33.23 | 8.91 | 0.00 | 44.89 |
| | FedAvg(1) | 6.63 | 1.13 | -63.71 | 6.79 |
| | PFNM | 6.89 | 6.00 | -64.61 | 8.73 |
| | Ensemble | 48.89 | 47.47 | -32.52 | 43.13 |
| | KD | 40.24 | 21.49 | -6.60 | 47.23 |
| | QKT | 61.08 | 57.16 | -21.55 | 48.69 |
| | QKT light | 58.44 | 58.87 | -28.98 | 46.27 |

Table 20: CIFAR10 (Path and Dir distributions)

| | | Acc | Query Acc. gain | Forgetting | Uniform Acc. |
|---|---|---|---|---|---|
| **Path** | **Single class queries** | | | | |
| | (Local: Acc = 31.91, Query class acc = 0.0) | | | | |
| | FedAvg | 53.12 | 59.50 | -15.34 | 49.76 |
| | FedProx | 53.63 | 59.70 | -14.79 | 50.26 |
| | Moon | 31.50 | 37.00 | -35.93 | 27.79 |
| | FT-FedAvg | 34.95 | 0.00 | -2.65 | 66.08 |
| | FedAvg(1) | 0.50 | 0.00 | -62.66 | 1.00 |
| | CLUE | 6.40 | 5.00 | -56.68 | 7.71 |
| | PFNM | 0.54 | 0.00 | -61.34 | 0.58 |
| | Ensemble | 34.83 | 34.30 | -29.50 | 36.16 |
| | KD | 39.86 | 30.20 | -14.10 | 48.42 |
| | QKT | 68.48 | 72.80 | -3.10 | 64.07 |
| | QKT light | 68.17 | 71.60 | -3.02 | 64.41 |
| | **Multi-class queries** | | | | |
| | (Local: Acc = 10.75, Query class acc = 0.0) | | | | |
| | FedAvg | 46.29 | 45.80 | -15.34 | 48.72 |
| | FedProx | 46.21 | 45.43 | -14.79 | 49.11 |
| | Moon | 22.44 | 21.01 | -35.93 | 26.42 |
| | FT-FedAvg | 12.50 | 0.59 | -2.65 | 52.94 |
| | FedAvg(1) | 1.11 | 1.11 | -62.66 | 1.11 |
| | PFNM | 1.70 | 1.68 | -61.07 | 1.50 |
| | Ensemble | 39.67 | 40.70 | -29.50 | 37.03 |
| | KD | 15.99 | 16.00 | -14.10 | 41.60 |
| | QKT | 54.56 | 54.56 | -10.18 | 55.06 |
| | QKT light | 54.59 | 54.16 | -10.37 | 54.87 |
| **Dir** | **Single class queries** | | | | |
| | (Local: Acc = 30.68, Query class acc = 02.89) | | | | |
| | FedAvg | 37.59 | 27.90 | -7.64 | 48.13 |
| | FedProx | 41.57 | 32.40 | -6.15 | 50.38 |
| | Moon | 19.09 | 10.60 | -16.03 | 26.61 |
| | FT-FedAvg | 34.19 | 5.40 | -3.93 | 36.69 |
| | FedAvg(1) | 0.51 | -2.90 | -35.31 | 1.50 |
| | CLUE | 0.97 | 0.00 | -35.25 | 0.66 |
| | PFNM | 0.37 | 0.00 | -35.94 | 0.87 |
| | Ensemble | 23.70 | 23.70 | -19.60 | 26.11 |
| | KD | 29.39 | 10.30 | -6.31 | 34.37 |
| | QKT | 51.42 | 43.99 | -3.36 | 33.39 |
| | QKT light | 49.37 | 41.30 | -3.83 | 30.98 |
| | **Multi-class queries** | | | | |
| | (Local: Acc = 13.08, Query class acc = 0.0) | | | | |
| | FedAvg | 43.99 | 40.83 | -7.64 | 47.89 |
| | FedProx | 47.78 | 44.70 | -6.15 | 50.25 |
| | Moon | 24.72 | 21.43 | -16.03 | 26.47 |
| | FT-FedAvg | 18.12 | 5.68 | -3.93 | 34.54 |
| | FedAvg(1) | 1.14 | -2.55 | -35.31 | 1.41 |
| | PFNM | 1.47 | 0.00 | -35.80 | 1.22 |
| | Ensemble | 30.76 | 27.67 | -19.60 | 26.56 |
| | KD | 25.37 | 17.47 | -6.31 | 33.51 |
| | QKT | 48.46 | 44.50 | -6.01 | 36.62 |
| | QKT light | 48.27 | 44.33 | -5.78 | 37.34 |

Table 21: CIFAR100 (Path and Dir distributions)

| | | Acc | Query Acc. gain | Forgetting | Uniform Acc. |
|---|---|---|---|---|---|
| **Path** | **Single class queries** | | | | |
| | (Local: Acc = 44.58, Query class acc = 0.0) | | | | |
| | FedAvg | 22.30 | 16.30 | -46.53 | 28.04 |
| | FedProx | 24.22 | 17.59 | -45.27 | 32.76 |
| | Moon | 40.99 | 44.22 | -37.37 | 43.90 |
| | FT-FedAvg | 45.55 | 0.00 | 0.00 | 61.58 |
| | FedAvg(1) | 8.87 | 10.00 | -69.37 | 10.00 |
| | CLUE | 26.23 | 38.78 | -63.98 | 22.22 |
| | PFNM | 19.27 | 21.61 | -68.47 | 14.13 |
| | Ensemble | 30.22 | 31.95 | -48.37 | 34.63 |
| | KD | 49.64 | 24.13 | -9.53 | 61.35 |
| | QKT | 71.02 | 82.68 | -22.02 | 65.42 |
| | QKT light | 71.27 | 82.70 | -21.67 | 65.71 |
| | **Multi-class queries** | | | | |
| | (Local: Acc = 27.04, Query class acc = 0.0) | | | | |
| | FedAvg | 27.95 | 27.44 | -46.53 | 29.71 |
| | FedProx | 27.56 | 23.17 | -45.27 | 32.41 |
| | Moon | 38.30 | 38.95 | -37.37 | 41.75 |
| | FT-FedAvg | 27.79 | 0.00 | 0.00 | 44.04 |
| | FedAvg(1) | 7.99 | 8.33 | -69.37 | 8.33 |
| | PFNM | 19.08 | 10.54 | -63.57 | 14.47 |
| | Ensemble | 26.31 | 23.75 | -48.37 | 30.98 |
| | KD | 39.53 | 23.91 | -10.69 | 50.93 |
| | QKT | 62.13 | 72.80 | -44.10 | 49.80 |
| | QKT light | 61.71 | 71.93 | -43.62 | 49.72 |
| **Dir** | **Single class queries** | | | | |
| | (Local: Acc = 42.65, Query class acc = 0.0) | | | | |
| | FedAvg | 30.39 | 18.52 | -31.67 | 47.61 |
| | FedProx | 32.31 | 20.12 | -30.71 | 47.38 |
| | Moon | 39.50 | 28.70 | -27.87 | 47.52 |
| | FT-FedAvg | 25.00 | 1.11 | 1.11 | 46.54 |
| | FedAvg(1) | 3.73 | 0.00 | -66.45 | 7.61 |
| | CLUE | 12.47 | 19.79 | -60.26 | 12.88 |
| | PFNM | 23.83 | 17.38 | -57.47 | 14.47 |
| | Ensemble | 43.61 | 37.36 | -32.14 | 45.30 |
| | KD | 50.64 | 27.81 | -10.61 | 50.45 |
| | QKT | 73.48 | 72.68 | -13.49 | 46.73 |
| | QKT light | 70.14 | 82.50 | -24.77 | 42.62 |
| | **Multi-class queries** | | | | |
| | (Local: Acc = 23.29, Query class acc = 0.0) | | | | |
| | FedAvg | 46.16 | 41.38 | -31.67 | 50.71 |
| | FedProx | 46.55 | 42.27 | -30.71 | 50.19 |
| | Moon | 46.16 | 43.16 | -27.87 | 47.88 |
| | FT-FedAvg | 25.00 | 1.11 | -1.36 | 37.64 |
| | FedAvg(1) | 10.64 | 9.17 | -66.45 | 10.79 |
| | PFNM | 12.75 | 13.64 | -61.05 | 10.68 |
| | Ensemble | 39.78 | 38.36 | -32.14 | 41.69 |
| | KD | 37.54 | 25.18 | -9.92 | 43.82 |
| | QKT | 54.57 | 55.38 | -24.93 | 43.58 |
| | QKT light | 51.01 | 59.07 | -38.21 | 38.72 |

Table 22: CINIC10 (Path and Dir distributions)

| | | Acc | Query Acc. gain | Forgetting | Uniform Acc. |
|---|---|---|---|---|---|
| **Path** | **Single class queries** | | | | |
| | (Local: Acc = 45.99, Query class acc = 0.0) | | | | |
| | FedAvg | 43.34 | 29.77 | -38.34 | 48.82 |
| | FedProx | 63.65 | 50.89 | -15.97 | 71.76 |
| | Moon | 72.97 | 69.63 | -16.74 | 75.78 |
| | FT-FedAvg | 46.99 | 0.00 | -2.88 | 70.45 |
| | FedAvg(1) | 14.89 | 10.00 | -76.96 | 12.50 |
| | PFNM | 8.85 | 10.00 | -81.49 | 10.00 |
| | CLUE | 20.67 | 20.00 | -74.16 | 17.50 |
| | Ensemble | 56.08 | 43.81 | -26.97 | 60.93 |
| | KD | 43.76 | 5.27 | -9.58 | 64.73 |
| | QKT | 77.65 | 88.42 | -24.71 | 71.91 |
| | QKT light | 78.14 | 88.03 | -24.35 | 72.55 |
| | **Multi-class queries** | | | | |
| | (Local: Acc = 31.62, Query class acc = 0.0) | | | | |
| | FedAvg | 53.33 | 51.25 | -38.34 | 53.26 |
| | FedProx | 67.94 | 63.31 | -15.97 | 72.39 |
| | Moon | 72.45 | 72.99 | -16.74 | 73.92 |
| | FT-FedAvg | 32.29 | 0.00 | -2.88 | 54.98 |
| | FedAvg(1) | 17.09 | 10.33 | -76.96 | 15.27 |
| | PFNM | 17.43 | 27.83 | -84.82 | 12.77 |
| | Ensemble | 61.11 | 57.21 | -26.97 | 62.58 |
| | KD | 38.04 | 14.58 | -9.58 | 56.05 |
| | QKT | 69.52 | 76.72 | -39.85 | 62.67 |
| | QKT light | 67.16 | 70.73 | -36.81 | 63.11 |
| **Dir** | **Single class queries** | | | | |
| | (Local: Acc = 57.20, Query class acc = 20.58) | | | | |
| | FedAvg | 63.13 | 35.79 | -11.95 | 70.51 |
| | FedProx | 72.75 | 47.30 | -7.93 | 77.96 |
| | Moon | 71.80 | 34.61 | -12.39 | 68.66 |
| | FT-FedAvg | 60.99 | 3.89 | -2.16 | 66.06 |
| | FedAvg(1) | 19.98 | 0.00 | -62.21 | 13.85 |
| | CLUE | 19.29 | 0.00 | -61.74 | 12.18 |
| | PFNM | 20.15 | 0.00 | -52.78 | 15.28 |
| | Ensemble | 69.35 | 43.84 | -18.90 | 65.04 |
| | KD | 56.40 | 23.26 | -13.58 | 62.30 |
| | QKT | 75.23 | 77.37 | -29.31 | 52.18 |
| | QKT light | 74.70 | 76.45 | -28.11 | 52.14 |
| | **Multi-class queries** | | | | |
| | (Local: Acc = 37.66, Query class acc = 18.19) | | | | |
| | FedAvg | 73.76 | 58.71 | -11.95 | 75.82 |
| | FedProx | 81.78 | 64.15 | -7.93 | 81.79 |
| | Moon | 71.40 | 56.25 | -12.39 | 73.73 |
| | FT-FedAvg | 45.28 | 7.97 | -2.16 | 55.95 |
| | FedAvg(1) | 14.73 | 0.00 | -62.21 | 13.45 |
| | PFNM | 15.49 | 0.00 | -52.74 | 15.48 |
| | Ensemble | 64.19 | 41.14 | -19.59 | 65.25 |
| | KD | 52.87 | 28.15 | -11.90 | 59.22 |
| | QKT | 63.31 | 55.11 | -33.31 | 54.16 |
| | QKT light | 64.26 | 56.16 | -31.87 | 55.53 |

Table 23: BloodMNIST (Path and Dir distributions)

| | | Acc | Query Acc. gain | Forgetting | Uniform Acc. |
|---|---|---|---|---|---|
| **Path** | **Single class queries** | | | | |
| | (Local: Acc = 43.61, Query class acc = 0.0) | | | | |
| | FedAvg | 49.88 | 60.88 | -43.80 | 49.12 |
| | FedProx | 62.12 | 76.11 | -33.22 | 60.11 |
| | Moon | 60.06 | 71.90 | -38.94 | 57.63 |
| | FT-FedAvg | 44.72 | 0.00 | -5.89 | 65.38 |
| | FedAvg(1) | 11.11 | 0.00 | -62.79 | 21.19 |
| | CLUE | 14.07 | 11.19 | -77.74 | 10.52 |
| | PFNM | 22.21 | 30.00 | -70.66 | 15.06 |
| | Ensemble | 60.03 | 74.55 | -41.80 | 53.49 |
| | KD | 58.13 | 32.08 | -8.66 | 70.14 |
| | QKT | 83.41 | 80.88 | -10.36 | 80.85 |
| | QKT light | 77.25 | 90.37 | -22.89 | 71.10 |
| | **Multi-class queries** | | | | |
| | (Local: Acc = 24.95, Query class acc = 0.0) | | | | |
| | FedAvg | 41.35 | 47.41 | -43.80 | 42.91 |
| | FedProx | 49.92 | 55.49 | -33.22 | 51.64 |
| | Moon | 47.16 | 51.00 | -38.94 | 49.34 |
| | FT-FedAvg | 25.87 | 0.00 | -5.89 | 45.78 |
| | FedAvg(1) | 21.12 | 23.30 | -62.79 | 25.77 |
| | PFNM | 16.43 | 21.03 | -70.74 | 12.58 |
| | Ensemble | 44.63 | 44.86 | -41.80 | 44.58 |
| | KD | 44.40 | 27.78 | -8.66 | 57.35 |
| | QKT | 67.03 | 71.28 | -38.76 | 62.88 |
| | QKT light | 50.62 | 55.97 | -45.38 | 47.38 |
| **Dir** | **Single class queries** | | | | |
| | (Local: Acc = 50.32, Query class acc = 11.21) | | | | |
| | FedAvg | 60.81 | 47.04 | -22.49 | 66.94 |
| | FedProx | 63.74 | 49.78 | -19.88 | 68.79 |
| | Moon | 67.44 | 56.11 | -16.15 | 69.98 |
| | FT-FedAvg | 55.28 | 5.89 | 0.00 | 56.40 |
| | FedAvg(1) | 3.05 | 0.00 | -67.31 | 11.65 |
| | CLUE | 11.96 | 0.00 | -64.46 | 8.31 |
| | PFNM | 11.90 | 10.14 | -69.73 | 12.16 |
| | Ensemble | 59.86 | 48.54 | -24.26 | 59.02 |
| | KD | 63.29 | 32.40 | -8.90 | 58.13 |
| | QKT | 77.95 | 72.56 | -14.95 | 54.54 |
| | QKT light | 78.13 | 73.71 | -15.36 | 55.29 |
| | **Multi-class queries** | | | | |
| | (Local: Acc = 34.08, Query class acc = 05.39) | | | | |
| | FedAvg | 57.51 | 44.86 | -22.49 | 66.58 |
| | FedProx | 59.85 | 47.82 | -19.88 | 67.94 |
| | Moon | 62.80 | 55.04 | -16.15 | 67.55 |
| | FT-FedAvg | 38.10 | 3.58 | 0.00 | 46.77 |
| | FedAvg(1) | 7.51 | 0.43 | -67.31 | 13.69 |
| | PFNM | 16.43 | 21.03 | -70.74 | 12.58 |
| | Ensemble | 60.37 | 56.08 | -24.26 | 61.87 |
| | KD | 52.84 | 33.93 | -8.90 | 53.37 |
| | QKT | 69.32 | 65.52 | -14.75 | 54.82 |
| | QKT light | 67.09 | 67.01 | -28.33 | 50.39 |

Table 24: PathMNIST (Path and Dir distributions)

