# OpenReview forum: "Query-based Knowledge Transfer for Heterogeneous Learning Environments"
_ICLR.cc/2025/Conference — ICLR 2025 Poster_

### Official Review · Reviewer_5p86 · 2024-11-02

**Soundness:** 2
**Presentation:** 3
**Contribution:** 2
**Rating:** 6
**Confidence:** 4

**Summary:**

The authors propose a framework called Query-based Knowledge Transfer (QKT). This framework uses a data-free masking strategy that facilitates more effective knowledge distillation, addressing statistically heterogeneous environments in Federated Learning.

**Strengths:**

-The paper is well-motivated. Figures 1 and 2 makes it easy to understand the strong motivations of this paper. The writing is also clear.
- The idea of using noise to identify expert models is interesting, as it avoids dodgy reliances of "public data" or "synthetic data" in federated learning context, where such data may be scarce.

**Weaknesses:**

-While the framework proposed seems general, it is not clear why only image-related tasks are considered.
- The accuracies on simple datasets in Table 2 seem surprisingly low, for the baseline approaches.
- The authors seem to have chosen some relatively general baselines to compare against, neglecting several Federated learning + knowledge distillation papers; such as Zhu, Z., Hong, J., & Zhou, J. (2021, July). Data-free knowledge distillation for heterogeneous federated learning. In International conference on machine learning (pp. 12878-12889). PMLR.
- the entire idea seems overly reliant on the overconfidence concept (Guo et al., 2017). This implies that a careful tuning of \lambda in (3) is needed. The experiments seem to not mention what happens when a client may be an expert in multiple classes at once, or when multiple clients are experts in a single class

**Questions:**

-how will the framework extend to other modalities?
- the authors claim that QKT completes learning in a single round. Is there convergence guarantees for this?
- using noise to check what expert a model is could also reveal the client's data distribution, which is itself a privacy intrusion

---

> ### Author Response · Authors · 2024-11-28
> **Response 1/3**
>
> We sincerely thank the reviewer for their thoughtful feedback and recognition of our work. We are pleased that you found the paper well-motivated, with clear writing and figures that effectively convey our objectives. Your acknowledgment of our innovative use of noise to identify expert models validates our approach and its practical value in addressing key challenges in collaborative learning.
> Here are our responses to the concerns.
>
>
> ### **1. Addressing Low Baseline Accuracies in Table 2**
> Thank you for highlighting this observation. The performance of baseline collaborative learning approaches under data heterogeneity in our experiments aligns with findings from prior works addressing similar scenarios. Specifically, federated learning with non-IID data distributions is known to result in performance degradation due to local data imbalance and limited diversity. Studies such as "No Fear of Heterogeneity" (NeurIPS 2021) and "Personalized Cross-Silo Federated Learning on Non-IID Data" (AAAI 2021) discuss how this heterogeneity leads to suboptimal performance for generalized and personalized FL methods.
> The baselines used in our experiments, such as FedAvg and FedProx, are tested under configurations comparable to those in prior research. Methods like PFNM and CLUE, designed for single-round learning, struggle with heterogeneity in data distributions and complex model structures, leading to suboptimal performance.
>
> This emphasizes the limitations of these methods in handling data heterogeneity effectively and highlights the need for targeted approaches like QKT, which addresses these challenges through query-specific, communication-efficient knowledge transfer.
>
>
> ### **2. Overconfidence, Tuning of λ, and Handling Multiple Expert Scenarios**
> Thank you for this insightful comment. We appreciate the opportunity to elaborate on these points.
>
> **Overconfidence and the Role of λ**: The concept of overconfidence is leveraged in our masking strategy to identify and filter irrelevant knowledge contributions. The parameter λ in Equation (3) controls the trade-off between focusing on query classes and retaining existing knowledge. While tuning λ is useful to balance this trade-off, our experiments show that λ values between 1.5 and 2 (we used 1.7 for most datasets) were effective, based on empirical observations from CIFAR-10. Even the default value of λ=1 demonstrated notable improvements, as shown in the ablation studies (Section 4.3). This indicates that QKT is not overly sensitive to λ under typical settings.
>
> **Clients as Experts in Multiple Classes**: When a teacher model is proficient in multiple classes, our approach naturally incorporates this by using the masking mechanism to retain knowledge relevant to all those classes. The synthetic Gaussian noise samples utilized during masking allow the model to effectively identify and focus on the teacher’s areas of expertise.
>
> **Multiple Clients as Experts in a Single Class**:
> In scenarios where multiple teachers are proficient in the same class, QKT aggregates knowledge from these teachers by combining their masked outputs, allowing the student to benefit from diverse perspectives while maintaining robustness.
>
> Our experiments evaluate both scenarios where a teacher is an expert in a single class (single-class query) and where a teacher is proficient in a subset of classes (multi-class query). In both cases, multiple teachers can be utilized simultaneously to achieve the student’s learning objectives, ensuring flexibility and robustness in real-world applications.
> We hope this response clarifies these aspects and demonstrates how QKT effectively handles such scenarios.

---

> ### Author Response · Authors · 2024-11-28
> **Response 2/3**
>
> ### **3. Comparisons with Relevant FL + KD Baselines**
> Thank you for highlighting the importance of including comparisons with relevant FL + KD baselines. In response to this and to a similar concern raised by another reviewer, we conducted additional experiments incorporating PerAda, a recent personalized FL baseline that integrates both KD and FL.
>
> These experiments were performed on CIFAR-10 using both pathological and Dirichlet data distributions, focusing on single-class and multi-class queries. To ensure a fair comparison, the shared parameters in PerAda were initialized by distilling knowledge from all local models. In this context, PerAda (local) refers to the average performance of personalized models trained individually for each client, while PerAda (global) reflects the average performance of the global model concerning each client’s query.
>
> Detailed results for PerAda are included in the table below, alongside additional baselines in Table 2 and Appendix A.7, Table 20. The results show that QKT consistently outperforms PerAda, achieving higher query accuracy gains, improved overall accuracy, and reduced forgetting rates. These results highlight QKT’s superior ability to address the challenges posed by heterogeneous data distributions and client-specific objectives in collaborative learning scenarios.
>
>
> ### Table 1: Pathological Distribution
>
> | Query Type              | Method           | Acc. | Query Acc. gain | Forgetting |
> |-------------------------|------------------|---------------|-----------------|------------|
> | **Single class queries**| FedAvg           | 0.52325       | 0.48260         | -0.31396   |
> |                         | FedProx          | 0.49685       | 0.45630         | -0.30280   |
> |                         | Moon             | 0.43172       | 0.44250         | -0.39923   |
> |                         | FT-FedAvg        | 0.46202       | 0.0             | 0.0        |
> |                         | PerAda (Local)   | 0.46205       | 0.00001         | 0.0        |
> |                         | PerAda (Global)  | 0.40428       | 0.34920         | -0.38526   |
> |                         | QKT              | 0.74557       | 0.77970         | -0.16810   |
> | **Multi-class queries** | FedAvg           | 0.51271       | 0.49957         | -0.31396   |
> |                         | FedProx          | 0.51896       | 0.50653         | -0.30280   |
> |                         | Moon             | 0.41582       | 0.41440         | -0.39923   |
> |                         | FT-FedAvg        | 0.28355       | 0.0             | 0.0        |
> |                         | PerAda (Local)   | 0.28272       | 0.0             | 0.0        |
> |                         | PerAda (Global)  | 0.44931       | 0.42263         | -0.38526   |
> |                         | QKT              | 0.65596       | 0.71396         | -0.32630   |
>
> ---
>
> ### Table 2: Dirichlet Distribution
>
> | Query Type              | Method           | Acc. | Query Acc. gain | Forgetting |
> |-------------------------|------------------|---------------|-----------------|------------|
> | **Single class queries**| FedAvg           | 0.60554       | 0.63800         | -0.21707   |
> |                         | FedProx          | 0.61146       | 0.63720         | -0.21054   |
> |                         | Moon             | 0.50280       | 0.54030         | -0.27476   |
> |                         | FT-FedAvg        | 0.49155       | 0.05660         | 0.0        |
> |                         | PerAda (Local)   | 0.44668       | 0.0             | 0.0        |
> |                         | PerAda (Global)  | 0.56696       | 0.54269         | -0.23021   |
> |                         | QKT              | 0.71346       | 0.71630         | -0.13520   |
> | **Multi-class queries** | FedAvg           | 0.59921       | 0.54784         | -0.21707   |
> |                         | FedProx          | 0.59514       | 0.53623         | -0.21054   |
> |                         | Moon             | 0.49458       | 0.44294         | -0.27476   |
> |                         | FT-FedAvg        | 0.33226       | 0.08913         | 0.0        |
> |                         | PerAda (Local)   | 0.25861       | 0.0             | 0.0        |
> |                         | PerAda (Global)  | 0.54384       | 0.46215         | -0.23021   |
> |                         | QKT              | 0.61076       | 0.57157         | -0.21546   |

---

> ### Author Response · Authors · 2024-11-28
> **Response 3/3**
>
> ### **4. Regarding Convergence Guarantees**
> Thank you for raising this insightful question. Convergence in the traditional sense of iterative optimization does not directly apply to QKT and, accordingly, we do not devise a theory for this. Instead, our empirical results across various datasets and scenarios demonstrate that QKT is effective in practice.
>
> As shown in Section 4.2 (Table 2), Section 4.3 (Table 3), and Appendix A.5 and A.6, QKT achieves stable and competitive performance after a single round of knowledge transfer and refinement. These results consistently highlight QKT's ability to balance learning the query task and mitigating forgetting, even under heterogeneous data distributions.
>
> While QKT’s empirical effectiveness is empirically supported, formal theoretical guarantees for convergence in decentralized and heterogeneous settings remain an open research area. Exploring such guarantees is a promising direction for future work, potentially building on theoretical frameworks from personalized federated learning and knowledge distillation. We will acknowledge this potential direction for future work.
> We hope this response clarifies QKT’s approach and its demonstrated ability to meet its learning objectives efficiently within a single round. Thank you for the opportunity to address this aspect.
>
>
>
>
>
> ### **5. Privacy Concerns with Noise-Based Filtering**
> Thank you for your comment regarding the potential privacy concerns of using noise to determine teacher expertise. In QKT, the synthetic noise-based filtering is designed to avoid revealing direct information about client data distributions. The generated noise inputs are independent of client-specific data, and the predictions reflect only the model’s inherent biases, as detailed in Section 3.3. These outputs are used solely for filtering irrelevant teachers and are not shared or used to infer other clients’ data.
>
> It is important to note that FL primarily aims to protect raw data from being shared. While sharing model weights or gradients is common, these can still be susceptible to privacy attacks such as membership inference (Shokri et al., CCS 2017), model inversion (Fredrikson et al., CCS 2015), and property inference (Melis et al., CCS 2019). However, general statistical information, such as the number of training samples, is often used in FL for model aggregation (McMahan et al., AISTATS 2017) or per-class statistics to handle heterogeneity (Wang et al., ICLR 2020).
>
> In highly privacy-sensitive scenarios, techniques like differential privacy (Abadi et al., CCS 2016), secure multi-party computation (Bonawitz et al., CCS 2017), or homomorphic encryption (Gentry, STOC 2009) could be investigated and potentially integrated into QKT.
>
> We hope this response clarifies how QKT aligns with FL privacy principles and its adaptability to stringent privacy requirements.

---

> > ### Comment · Reviewer_5p86 · 2024-12-01
> >
> > I am satisfied with the rebuttal. I will raise my score to 6.

---

### Official Review · Reviewer_2vF7 · 2024-11-04

**Soundness:** 2
**Presentation:** 2
**Contribution:** 1
**Rating:** 6
**Confidence:** 3

**Summary:**

The paper introduces Query-based Knowledge Transfer (QKT) to learn query class in a collaborative learning setting while preventing interference from irrelevant information and mitigating catastrophic forgetting. It achieves this by implementing a two-phase distillation process that involves query-focused learning and classification head refinement.
Experimental results demonstrate that QKT outperforms traditional federated learning and other collaborative methods in single- and multi-class query tasks.

**Strengths:**

- The variant QKT Light can reduce communication overhead by requiring only one communication round, making it well-suited for communication-limited settings.
- The authors demonstrate the adaptability of the proposed method across diverse datasets and tasks, including medical and standard image classification benchmarks.

**Weaknesses:**

- The practical applications or user cases motivating this approach for query-based knowledge transfer are not clearly defined, especially given that personalized federated learning can also handle similar scenarios.
- The novelty of the paper's contributions is somewhat limited, as knowledge distillation and classification head refinement are established methods in personalized FL​  [1,2,3,4]
- Table 2 lacks comparisons with advanced personalized FL baselines after 2022 such as those in [5,6].
- Table 1 also lacks a baseline of standalone local fine-tuning on the query class, which is the simplest way to learn the query class without any communication. How effective is this baseline?
- QKT's performance is highly sensitive to the hyperparameters (e.g., $\lambda$ in the mask shown in Table 4), which could be challenging to optimize in real-world scenarios.


reference:
- [1] Data-Free Knowledge Distillation for Heterogeneous Federated Learning  ICML 2021
- [2] On bridging genertic and personalized federated learning for image classification. ICLR 2022
- [3] Think locally, act globally: Federated learning with local and global representations. 2020
- [4] Exploiting shared representations for personalized federated learning ICML 2021
- [5] Test-time robust personalization for federated learning. ICLR, 2023.
- [6] PerAda: Parameter-Efficient Federated Learning Personalization with Generalization Guarantees CVPR 2024

**Questions:**

- It is unclear why learning irrelevant classes is inherently problematic, as shown in Figure 1; incorporating such classes could potentially enhance the model’s ability to handle out-of-distribution data in test time [5].
- In Phase 1 (lines 219-221), it seems to be assumed that clients are aware of the irrelevant classes for defining the threshold $\tau$. Does it mean that each client knows the complete set of available classes [C]? This might raise questions about the method's applicability in real-world decentralized settings where it could be hard to know what other clients’ classes are.


post-rebuttal: Thanks to the authors for addressing my concerns about motivation and baselines. I raised the score accordingly.

---

> ### Author Response · Authors · 2024-11-28
> **Response 1/3**
>
> We sincerely thank the reviewer for their feedback.  We are pleased that the efficiency of our approach in reducing communication overhead and its adaptability across diverse datasets and tasks were appreciated. Here are our responses to the concerns.
>
> ### **1. Motivation and Practical Applications of QKT**
> Thank you for this insightful comment. We appreciate the opportunity to clarify the practical applications that motivate QKT and how it differs from personalized federated learning (PFL). We have included the motivations in lines 34–39 and expand on them here.
>
> **Motivation and Practical Applications**:
> QKT is particularly suited for scenarios where clients or entities own specialized knowledge that can enhance the performance of other clients with similar needs, even if the required knowledge is limited or absent in their local data. For instance, in healthcare, hospitals may need specific expertise on rare or emerging diseases not represented in their local datasets. With QKT, these hospitals can benefit from the relevant knowledge of peer institutions with similar cases, enhancing diagnostic capabilities without sharing sensitive data. Other practical applications include finance, where institutions may need insights on regional fraud patterns, and cybersecurity, where organizations require knowledge of emerging threats observed in other sectors.
>
> **Distinction from Personalized FL**:
> While PFL aims to tailor models to individual clients, it typically relies on iterative, collaborative training rounds and may not effectively address cases where critical knowledge is limited or entirely absent in a client’s local data. QKT fills this gap by enabling direct knowledge transfer from pre-trained peers to clients needing specific information, without requiring iterative rounds. This approach provides a scalable, efficient solution for cases requiring targeted knowledge adaptation, especially when immediate integration of specialized expertise is essential.
>
>
>
>
>
> ### **2. Novelty of QKT Contributions**
> Thank you for this comment. We acknowledge that both knowledge distillation and classification head refinement have been applied in various forms within personalized FL, as indicated in recent works. However, FL is not a well-suited solution to our problem, which we describe below.
>
> Novelty of QKT:
> QKT introduces a query-based, selective knowledge transfer mechanism that addresses scenarios where critical knowledge may be limited or absent in a client’s local data. QKT directly enables a client to acquire targeted knowledge from multiple peer models that are already pre-trained on relevant tasks, without the need for iterative communication rounds. Additionally, our approach leverages a data-free masking strategy to focus the knowledge transfer only on relevant classes from appropriately selected teacher models. This selective transfer reduces interference from irrelevant knowledge, enhancing task-specific model performance in a way that standard PFL methods may not support, as they typically rely on local training rather than targeted, query-specific adaptation.

---

> > ### Author Response · Authors · 2024-11-28
> > **Response 2/3**
> >
> > ### **3. Comparisons with Advanced Baselines**
> > Thank you for your feedback. In the original manuscript, we included the simple yet competitive personalized FL approach FT-FedAvg to highlight the general limitations of personalized FL methods in addressing our primary objective: enabling clients to improve performance on specific query classes that are absent or under-represented in their local data. Additionally, we demonstrated the shortage of generalized FL approaches to meet this objective under heterogeneous data distributions.
> >
> > In response to your comment, we conducted additional experiments comparing QKT with PerAda, a recent personalized FL baseline [6]. These experiments use CIFAR-10 under pathological and Dirichlet distributions, focusing on single-class and multi-class queries and following the experimental setup explained in Section 4. For fair comparison, the shared parameters in PerAda were initialized by combining all local models through knowledge distillation. The results are summarized in the table below. PerAda (local) represents the average performance of personalized models trained for each client, while PerAda (global) represents the average performance of the global model with respect to each client's query. Additional baselines are in Table 2 and Appendix A.7, Table 20.
> >
> > PerAda demonstrates competitive performance compared to other FL approaches, particularly with its global adapter mechanism, aligning with its emphasis on parameter efficiency and generalization. However, QKT consistently achieves higher query accuracy gains, better overall accuracy, and lower forgetting rates, demonstrating its superior ability to address heterogeneous data and client-specific objectives.
> >
> > ### Table 1: Pathological Distribution
> >
> > | Query Type              | Method           | Acc. | Query Acc. gain | Forgetting |
> > |-------------------------|------------------|---------------|-----------------|------------|
> > | **Single class queries**| FedAvg           | 0.52325       | 0.48260         | -0.31396   |
> > |                         | FedProx          | 0.49685       | 0.45630         | -0.30280   |
> > |                         | Moon             | 0.43172       | 0.44250         | -0.39923   |
> > |                         | FT-FedAvg        | 0.46202       | 0.0             | 0.0        |
> > |                         | PerAda (Local)   | 0.46205       | 0.00001         | 0.0        |
> > |                         | PerAda (Global)  | 0.40428       | 0.34920         | -0.38526   |
> > |                         | QKT              | 0.74557       | 0.77970         | -0.16810   |
> > | **Multi-class queries** | FedAvg           | 0.51271       | 0.49957         | -0.31396   |
> > |                         | FedProx          | 0.51896       | 0.50653         | -0.30280   |
> > |                         | Moon             | 0.41582       | 0.41440         | -0.39923   |
> > |                         | FT-FedAvg        | 0.28355       | 0.0             | 0.0        |
> > |                         | PerAda (Local)   | 0.28272       | 0.0             | 0.0        |
> > |                         | PerAda (Global)  | 0.44931       | 0.42263         | -0.38526   |
> > |                         | QKT              | 0.65596       | 0.71396         | -0.32630   |
> >
> > ---
> >
> > ### Table 2: Dirichlet Distribution
> >
> > | Query Type              | Method           | Acc. | Query Acc. gain | Forgetting |
> > |-------------------------|------------------|---------------|-----------------|------------|
> > | **Single class queries**| FedAvg           | 0.60554       | 0.63800         | -0.21707   |
> > |                         | FedProx          | 0.61146       | 0.63720         | -0.21054   |
> > |                         | Moon             | 0.50280       | 0.54030         | -0.27476   |
> > |                         | FT-FedAvg        | 0.49155       | 0.05660         | 0.0        |
> > |                         | PerAda (Local)   | 0.44668       | 0.0             | 0.0        |
> > |                         | PerAda (Global)  | 0.56696       | 0.54269         | -0.23021   |
> > |                         | QKT              | 0.71346       | 0.71630         | -0.13520   |
> > | **Multi-class queries** | FedAvg           | 0.59921       | 0.54784         | -0.21707   |
> > |                         | FedProx          | 0.59514       | 0.53623         | -0.21054   |
> > |                         | Moon             | 0.49458       | 0.44294         | -0.27476   |
> > |                         | FT-FedAvg        | 0.33226       | 0.08913         | 0.0        |
> > |                         | PerAda (Local)   | 0.25861       | 0.0             | 0.0        |
> > |                         | PerAda (Global)  | 0.54384       | 0.46215         | -0.23021   |
> > |                         | QKT              | 0.61076       | 0.57157         | -0.21546   |

---

> > > ### Author Response · Authors · 2024-11-28
> > > **Response 3/3**
> > >
> > > ### **4. Effectiveness of Standalone Local Fine-Tuning Baseline**
> > > Thank you for this comment. In the manuscript, we introduced the Query Accuracy Gain (Query Acc. Gain) metric, described in Section 4 (lines 360–361) and Appendix A.1. This metric effectively measures the improvement in query class accuracy before and after knowledge transfer. Results for this metric are illustrated in Figure 4 and expanded in Appendix A.7.
> > >
> > > In response to your comment, we have now included the standalone local fine-tuning baseline in Appendix A.7. This baseline evaluates the performance of clients’ models when fine-tuned solely on their local data, without leveraging any external knowledge transfer or communication. As described in Section 4 (lines 351–352), the queries are selected from classes that are absent or underrepresented in the client’s local data.
> > >
> > >
> > >
> > > ### **5. Sensitivity to Hyperparameters**
> > > Thank you for highlighting the sensitivity of QKT's performance to the hyperparameter λ. We hope to clarify that although λ is a useful parameter in our algorithm, we did not spend much time tuning this parameter.
> > >
> > > The parameter λ in QKT is useful for balancing the trade-offs between focusing on query classes and retaining knowledge of existing classes. It essentially controls the intensity of knowledge distillation, helping to prevent catastrophic forgetting while enhancing the model's ability to learn new, task-specific information.
> > >
> > > In our experiments, we utilized a λ ranging between 1.5 and 2, specifically 1.7 for most datasets based on empirical observations from CIFAR-10. This range was selected to facilitate performance improvements while maintaining an acceptable level of forgetting. Notably, even the default value of 1 demonstrated notable improvement across various tasks, indicating the robustness of the QKT approach, as shown in the ablation studies (Section 4.3).
> > >
> > > Although tuning λ requires additional experimentation, the process is straightforward. The parameter can be incrementally increased as long as forgetting remains within an acceptable range, depending on the task requirements. For example, in CIFAR-100, we observed that QKT consistently achieved significantly less forgetting than the baselines. This allowed us to increase λ, placing more emphasis on query classes while still outperforming the baselines in terms of forgetting.
> > >
> > >
> > >
> > > ### **6. Handling Irrelevant Classes in QKT**
> > > Incorporating irrelevant classes can potentially benefit out-of-distribution robustness, as noted in [5]. However, QKT prioritizes improving accuracy for targeted query classes while preserving local knowledge. Irrelevant classes may introduce interference, reducing query and local class performance. Extending QKT to enhance out-of-distribution robustness is an exciting direction for future work.
> > >
> > >
> > >
> > > ### **7. Applicability in Decentralized Settings**
> > > In Phase 1, the threshold τ is used to filter irrelevant classes by identifying teacher models proficient in query classes. It is important to note that QKT does not assume that each client knows the complete set of classes available across all clients.
> > >
> > > Instead, the relevant information—i.e., which classes a teacher is proficient in—is inferred locally by the student based on the teachers' responses to synthetic noise inputs, as explained in Section 3.3. This process involves using Gaussian noise samples to estimate the expertise of teacher models for specific classes without requiring prior knowledge of other clients' data distributions. Therefore, the applicability of QKT is not hindered by the decentralized nature of real-world settings where clients lack visibility into others' datasets.
> > >
> > >
> > >
> > > We hope these clarifications address your concerns and further emphasize the contributions and strengths of QKT. Thank you again for your valuable feedback.

---

> > > > ### Comment · Reviewer_2vF7 · 2024-12-02
> > > >
> > > > Thank you to the authors for the detailed clarifications and results! I have a few remaining questions and comments:
> > > >
> > > > Q1: The authors mentioned that the proposed algorithm is designed to address specific scenarios that traditional FL cannot handle, such as rare or emerging diseases in healthcare and regional fraud patterns in finance. However, I didn’t see a direct evaluation of these scenarios in the experiments. Did I overlook something?
> > > >
> > > > Q3: Could the authors clarify how the hyperparameters for the baseline methods used in Figure 4 were selected?
> > > >
> > > > Q4: In Appendix A.7, which row in the tables corresponds to standalone local fine-tuning?
> > > >
> > > > Q7: Do the clients have prior knowledge of the total number of classes $C$?

---

> ### Author Response · Authors · 2024-12-02
> **Follow-Up to Reviewer 2vF7**
>
> We wanted to express our gratitude once again to the reviewer for their valuable feedback and for raising critical points about our work. We hope that the detailed responses and additional experiments we provided have clarified the motivations behind QKT, its distinction from personalized FL, and the innovations introduced by our approach, including query-specific knowledge transfer, data-free masking, and the two-stage framework.
>
> In response to your comments, we also included additional baselines, such as PerAda and standalone local fine-tuning, as well as expanded results in Appendix A.7. These experiments consistently show that QKT outperforms alternative methods across key metrics, demonstrating its ability to handle query-based objectives and heterogeneous data effectively.
>
> If there are any remaining concerns or further clarifications needed, we’d be happy to address them before the discussion period concludes. We also kindly invite you to reconsider your score in light of the evidence and improvements we’ve presented.
>
> Thank you again for your time and thoughtful feedback.

---

> ### Author Response · Authors · 2024-12-02
> **Response: Follow-Up Questions and Comments**
>
> We sincerely thank the reviewer for their detailed follow-up questions and for engaging with our work. Here are our responses to the questions.
>
> **Q1: The authors mentioned that the proposed algorithm is designed to address specific scenarios that traditional FL cannot handle, such as rare or emerging diseases in healthcare and regional fraud patterns in finance. However, I didn’t see a direct evaluation of these scenarios in the experiments. Did I overlook something?**
>
> Thank you for your question. While we do not explicitly evaluate rare diseases or fraud patterns, our experiments simulate analogous challenges.
>
> Specifically, we utilize heterogeneous data distributions, including pathological and Dirichlet distributions, to mimic real-world scenarios where certain classes are under-represented or absent locally. These distributions align with challenges in applications like rare diseases or regional fraud patterns, where specific knowledge is unevenly distributed among clients. This experimental setup aligns with prior work in decentralized learning (e.g., McMahan et al., 2017; Fallah et al., 2020) to ensure fair comparisons, as described in Section 4.
>
> Additionally, we included medical benchmarks such as PathMNIST and BloodMNIST to evaluate QKT’s applicability in domains like healthcare. PathMNIST comprises colorectal cancer images, and BloodMNIST features blood cell microscopy images. These datasets represent realistic settings where knowledge transfer can enhance performance on rare or specialized tasks without direct data sharing.
>
> While the current experiments simulate these dynamics, we acknowledge the value of more direct evaluations in such domains. Extending QKT to explicitly target these application areas represents an exciting direction for future work.
>
> ---
>
> **Q3: Could the authors clarify how the hyperparameters for the baseline methods used in Figure 4 were selected?**
>
> Thank you for your question regarding the hyperparameters for the baseline methods in Figure 4. Detailed information about the hyperparameters and training settings is provided in Appendix A.1 (lines 616–635) of the manuscript.
>
> We maintained consistency in the training setup (e.g., optimizer, learning rate) across all methods to ensure fair comparisons. The other hyperparameters used for each baseline generally follow the values recommended in their respective original papers.
>
> ---
>
> **Q4: In Appendix A.7, which row in the tables corresponds to standalone local fine-tuning?**
>
> Thank you for your question regarding standalone local fine-tuning in Appendix A.7.
>
> The Standalone Local Fine-Tuning baseline corresponds to the rows labeled as "Local" in the tables. These rows represent client performance without external knowledge transfer and serve as a reference for evaluating query accuracy gains.
>
> Note that the metrics "Query Accuracy Gain" and "Forgetting" are not applicable to this baseline, as it does not involve any knowledge transfer process.
>
> ---
>
> **Q7: Do the clients have prior knowledge of the total number of classes \(C\)?**
>
> Thank you for this insightful question. QKT does not require clients to have explicit prior knowledge of the total number of classes across all clients. Instead, during Phase 1, clients infer relevant class information locally by analyzing teacher models’ responses to Gaussian noise inputs, as described in Section 3.3.
>
> It is important to clarify that FL and collaborative learning frameworks typically assume implicit consistency in the number and order of classes across clients. This standard assumption in FL (e.g., McMahan et al., AISTATS 2017) ensures that models trained collaboratively or shared among clients can interpret class labels consistently. Under this assumption, clients know that the class set is consistent globally but do not necessarily need explicit knowledge of how all other clients' data is distributed or labeled.
>
> In the context of QKT, this assumption allows clients to operate within their local knowledge while dynamically inferring class relevance through interactions with teacher models.
>
> That said, we acknowledge that some real-world scenarios may deviate from this assumption—such as tasks where class definitions or orders vary dynamically across clients. Addressing such challenges would involve mechanisms like label mapping or ontology alignment, which represent exciting opportunities for future work.
>
> We hope this clarification addresses your concerns and reinforces QKT’s adherence to established FL practices.
>
> ---
>
> **Remark**
>
> We deeply appreciate the reviewer’s engagement and thoughtful questions, which have significantly enriched our discussions and strengthened the manuscript. If further clarifications are required, we are happy to address them promptly before the discussion period concludes. We kindly invite you to reconsider your score based on the evidence and refinements we have provided.

---

### Official Review · Reviewer_PBtB · 2024-11-06

**Soundness:** 2
**Presentation:** 2
**Contribution:** 2
**Rating:** 5
**Confidence:** 3

**Summary:**

The paper introduces a framework that addresses the challenges arising from knowledge interference and catastrophic forgetting in FL due to the collaboration mechanisms that transfer knowledge between clients through weights. This is achieved by a data-free masking strategy that masks "irrelevant" knowledge from peer clients and a two-phase training process to balance knowledge transfer and retention.

**Strengths:**

1. The problem defined in the paper is interesting and the problems of knowledge interference and catastrophic forgetting make sense.
2. The paper is easy to understand.

**Weaknesses:**

1. Some of the claims like "QKT performs knowledge transfer in a single round, which is significantly more efficient than traditional centralized and peer-to-peer FL approaches that involve multiple communication rounds.", do not seem well justified since the goal of FL is joint collaborative training vs goal of this work is to distill knowledge from well trained peers.
2. The algorithmic details are not well presented - for example, how the teacher models' are trained and how their predictions are sent to the student model to obtain loss in Eq 4 are missing.
3. Some algorithmic details like how communication rounds = 100 were set for other approaches and what synthetic datasets were used are also missing.

**Questions:**

1. How would the methods compare in more homogeneous data settings? And how does the quality of the teacher model impact the performance? Is there a criterion to eliminate teacher models apart from the class based criterion.
2. Can you elaborate on what practical use case would this method be helpful in? And how does this work compare with body of work in knowledge distillation from multiple teachers in centralized settings?

---

> ### Author Response · Authors · 2024-11-26
> **Response 1/2**
>
> We sincerely thank the reviewer for their feedback. We're glad that our proposed decentralized collaborative learning problem is appreciated. Here are our responses to the concerns.
>
> ### **1. Some of the claims do not seem well justified**
> Thank you for this insightful comment. We agree that the goals of QKT and FL differ, and we are happy to reword that sentence toning down the claim. However, we would like to highlight that this sentence is in the context of a general discussion on efficiency considerations and we point out that QKT by design aims to minimize communication volumes.
>
>
> ### **2. The algorithmic details are not well presented**
> Thank you for highlighting this point. We are pleased to clarify the algorithmic details regarding teacher model training and the knowledge transfer process in QKT.
>
> Each teacher model is independently pre-trained on its local data, as detailed in Appendix A.1. As outlined in Section 3.3, QKT adopts an approach similar to peer-to-peer federated learning, in which the student client receives the relevant teacher models (the teacher set as defined in line 218) and then applies knowledge transfer locally. This single-round transfer reduces communication overhead and avoids iterative communication rounds.
>
> The KD loss in Equation 4 is computed locally by combining cross-entropy and KL divergence, both calculated on the student’s dataset. The KL divergence term is derived from the filtered teacher predictions, ensuring that only relevant knowledge from each teacher contributes to the KD process. Our data-free masking approach further enhances this relevance by filtering out unnecessary information, keeping the student’s focus task-specific.
>
> We hope this explanation clarifies the training and knowledge transfer mechanism in QKT.
>
> ### **3. how communication rounds were set for other approaches and what synthetic datasets were used**
> Thank you for raising this point. We clarify these aspects below:
>
> Communication Rounds: For FL baselines such as FedAvg, FedProx, and Moon, we use 100 communication rounds, a common choice in federated learning, aligning with standard practices and allowing for fair comparisons across methods (e.g., MOON, CVPR 2021; FedAPEN, NeurIPS 2023; Cross-Silo Prototypical Calibration, ICML 2023).
>
> Synthetic Dataset for Masking: As outlined in Section 3.3 and examined further in Section 4.3, our masking approach leverages Gaussian noise (we used 20 samples) shaped to match the model’s input dimensions. This synthetic noise enables the identification of relevant classes for knowledge transfer, ensuring that only essential information is utilized by the student model.
>
> We hope this response clarifies our chosen settings and the role of synthetic data in QKT’s masking strategy.
>
> ### **4. methods in more homogeneous data settings**
> We have experimented with the various levels of heterogeneity in Appendix A2. We further elaborate on this point here. To evaluate the impact of data homogeneity, we conducted experiments across varying levels of heterogeneity, from highly heterogeneous (M=2) to increasingly homogeneous (M=6) settings under the pathological distribution using CIFAR-10. As data becomes more homogeneous (higher M), the performance gap between QKT and other methods decreases. However, QKT consistently outperforms other approaches across all levels of heterogeneity.
>
> | Queries               | Method   | M=2       | M=3       | M=4       | M=5       | M=6       |
> |-----------------------|----------|-----------|-----------|-----------|-----------|-----------|
> | **Single class queries** |          |           |           |           |           |           |
> |                       | FedAvg   | 0.1228558 | 0.523251  | 0.5396444 | 0.6428220 | 0.75866965|
> |                       | Ensemble | 0.160359  | 0.24844   | 0.491168346| 0.6630863 | 0.6827722 |
> |                       | KD       | 0.39918883| 0.5197435 | 0.51624245| 0.60713961| 0.6487054 |
> |                       | QKT      | **0.5472083** | **0.74557** | **0.776483491** | **0.7873129** | **0.783894547** |
> | **Multi-class queries** |          |           |           |           |           |           |
> |                       | FedAvg   | 0.1263395 | 0.512171  | 0.51966131| 0.6672241 | **0.7326506** |
> |                       | Ensemble | 0.25124510| 0.37377   | 0.5747512 | 0.567808  | 0.6726978 |
> |                       | KD       | 0.22989422| 0.4303562 | 0.4498336 | 0.54866111| 0.5691662 |
> |                       | QKT      | **0.3372064** | **0.6559604** | **0.6483592** | **0.6999377** | **0.7323124** |

---

> ### Author Response · Authors · 2024-11-26
> **Response 2/2**
>
> ### **5. Impact of Teacher Model Quality**
> Teacher quality directly affects knowledge transfer. This is addressed in our experiments. In Section 4.3 (Table 3), we show that using all teachers in KD introduces noise, which negatively impacts performance. QKT’s filtering mechanism effectively mitigates this by focusing on relevant teachers, significantly improving the accuracy gain of KD.
>
> While QKT’s data-free filtering criterion prioritizes teachers with knowledge relevant to the query task, its masking approach further mitigates the risks posed by weaker teachers. This allows QKT to extract valuable specialized knowledge even from less reliable teachers while minimizing the impact of irrelevant or noisy contributions.
>
> Additionally, in larger-scale experiments with more clients (Appendix A.7), we compare KD with random participant selection to QKT, which incorporates teacher filtering. The results demonstrate that QKT consistently outperforms by effectively identifying and leveraging relevant teachers, even as the number of clients increases.
>
> These findings highlight the importance of teacher quality and demonstrate QKT’s ability to ensure reliable and efficient knowledge transfer across diverse and large-scale learning settings.
>
> ### **6. Teacher Selection Criterion**
> QKT currently only employs noise-based filtering to decide whether to select a teacher model or not. Sec 4.3 provides a corroboration for this choice. We will highlight that future work can consider expanding the criteria for teacher selection. For instance, teachers could be scored based on additional factors, such as the similarity between their predictions and the student model’s predictions. High-scoring teachers, reflecting greater task relevance and alignment, could then be prioritized for communication, ensuring scalability, efficiency, and reduced risk of forgetting previously acquired knowledge.
>
> ### **7. practical use case; difference with knowledge distillation from multiple teachers in centralized settings**
>
> Thank you for your question. QKT is particularly useful in scenarios where different clients or entities possess specialized knowledge that can be leveraged to enhance the performance of other clients with similar needs, as mentioned in lines 036-039. Practical use cases include healthcare, where hospitals could benefit from sharing expertise on rare or emerging diseases without exposing patient data, and autonomous driving, where one system could learn from diverse geographic and environmental data collected by others.
>
> In contrast to centralized settings, QKT operates in a decentralized manner, transferring knowledge directly from multiple pre-trained peer models to a student model without requiring central aggregation. This design is particularly advantageous in environments where data centralization is impractical or restricted due to privacy regulations.
>
> Furthermore, QKT does not rely on an additional transfer set, which is often required for knowledge distillation. Such datasets may not be readily available and, if shared or centralized, could violate privacy policies. Instead, QKT leverages the student’s own local data for knowledge transfer, ensuring privacy compliance while maintaining effectiveness.
>
> By enabling the integration of specialized knowledge from multiple sources, QKT allows the student model to quickly adapt to specific tasks. This makes QKT a scalable, flexible, and privacy-preserving solution for distributed learning environments.

---

> > ### Comment · Reviewer_PBtB · 2024-12-02
> > **Response to authors' rebuttal**
> >
> > I thank the authors for their rebuttal. I have updated my score in response to the rebuttal.
> >
> > Thank you!

---

### Official Review · Reviewer_mLpA · 2024-11-08

**Soundness:** 2
**Presentation:** 2
**Contribution:** 2
**Rating:** 6
**Confidence:** 3

**Summary:**

This paper addresses a challenge where the models are learned from heterogeneous environments. In specific, different clients can train their models with their local data. To this end, the authors introduce QKT to enhance the local model (a.k.a. student model) by distilling the knowledge from other pretrained models (a.k.a. teacher models). The solution consists of two phases: Feature Extractor Enhancement and Classification Head Refinement. The first phase filters out the irrelevant teacher models and uses them to facilitate the performance of the student model, and the second phase takes a further step to train the classifier. Extensive experiments demonstrate the effectiveness of the proposed method.

**Strengths:**

1. This paper proposes a new decentralized collaborative learning approach to enhance the performance of a client.
2. This work is well-motivated by identifying the limitations of knowledge distillation.
3. The work is verified on various datasets and demonstrates superior performance over other baselines.

**Weaknesses:**

1. The authors should discuss the recent development of ensemble learning rather than (personalized) federated learning. The proposed QKT relies on a number of pretrained teacher models and looks pretty similar to ensemble learning settings.
2. The proposed work requires pretraining for numerous teacher models. I cannot find the pretraining details in the paper. The authors should discuss this issue. Moreover, since these models store on a client, it leads to a significant storage burden if the client does not have sufficient storage/computation resources.

**Questions:**

See **Weaknesses**.

---

> ### Author Response · Authors · 2024-11-25
> **Response**
>
> We sincerely thank the reviewer for their feedback. We're glad that our proposed decentralized collaborative learning approach, its strong motivation, and its superior performance across diverse datasets compared to baselines were appreciated. Here are our responses to the concerns.
>
> ### **1. The authors should discuss the recent development of ensemble learning**
> Thank you for this thoughtful comment. We agree that ensemble learning shares some conceptual similarities with QKT, particularly in using multiple teacher models to improve the learning process. However, QKT diverges from traditional ensemble learning in several ways:
> - In this work, we focus on the problem of transferring knowledge from models trained on heterogeneous distributions, which aligns closely with personalized federated learning. Ensemble learning methods, such as deep ensembles (Lakshminarayanan et al., NeurIPS 2017) and snapshot ensembles (Huang et al., ICLR 2017), typically aggregate outputs from multiple models to improve prediction performance. However, applying such mechanisms to models trained with unique objectives or on heterogeneous data distributions, as seen in federated learning, remains an open challenge (Zhang et al., ICML 2021). Additionally, ensemble methods may still face knowledge inference challenges outlined in Section 3.1.
> - In our main experiment, we included a basic ensemble baseline (line 315, table 2) to demonstrate how ensemble methods yield unsatisfactory results (typically, with a large gap) in meeting clients' unique requirements in heterogeneous learning environments.
> - Besides, ensemble methods often require multiple forward passes and increased storage during inference, as predictions are combined across models. In contrast, QKT consolidates knowledge into a single student model, reducing both inference time and storage costs.
>
> ### **2. The authors should discuss the pre-training.  Moreover, it leads to a significant storage burden if the client does not have sufficient storage/computation resources.**
> Thank you for raising this important point. Pretraining is indeed an integral part of our setup. As detailed in Appendix A.1, each teacher model undergoes independent pretraining on its local dataset for up to 100 epochs, with early stopping applied if validation performance stagnates.
>
> It is indeed common in the collaborative training setting, as targeted by many federated learning methods, for each client to participate in training and to be able to locally train its own copy of the model on its local dataset. Our work makes similar assumptions regarding the ability of clients to store and train locally.
>
> Moreover,  QKT introduces significant improvements in resource efficiency compared to other collaborative learning approaches. Unlike peer-to-peer collaborative learning methods, which typically require multiple rounds of communication and storage of intermediate models, QKT performs knowledge transfer in a single communication round. This minimizes both computational and temporary storage demands at the client level. By consolidating relevant knowledge into a single student model, QKT also reduces the storage burden associated with ensemble methods, which would otherwise require maintaining and evaluating multiple models during inference.
>
> In addition, our data-free filtering approach could be integrated into peer-to-peer federated learning frameworks to further reduce storage and communication overhead. Typically, peer-to-peer personalized frameworks require all-to-all communication to identify suitable collaborator clients for each task. By adopting our data-free filtering approach, instead of transmitting (and temporarily storing) all models to each client—a process that could incur significant storage burdens—clients could instead share only the predictions from Gaussian noise samples. Each client can then determine suitable collaborators by selecting clients with similar tasks based on these predictions, ensuring more efficient, targeted communication with lower storage requirements.
>
> We hope this clarifies our approach and addresses your concerns regarding pretraining and resource efficiency.

---

> > ### Comment · Reviewer_mLpA · 2024-11-27
> >
> > Thank the authors for their responses. I think you may misunderstand the second weakness. Lines 209 --- 218 mention that a client should load multiple teacher models. Therefore, the server should transmit several teacher models to the client, and then the client preserves the suitable teacher models for training. During training, the client not only loads her local model $\theta_s$ but also incorporates the teacher set {$\theta\_t$}$\_{t \in \mathcal{T}_\mathcal{Q}}$ for calculating the logits of the sampled data. The client's computation resources (e.g., GPU) may not support to load all these models simultaneously. The authors should discuss how this issue should be addressed.
> >
> > Moreover, Eq. (2) sounds weird to some extent. The authors claim that a well-trained teacher model has a higher probability on the trained classes when the input sample is a Gaussian noise. Consider a binary classification task where the teacher is always trained with single-class data. If a data point matches the only class, the teacher model will likely return a high probability on that class. Otherwise, the output should be random because the teacher model never sees such a data point. The authors should justify why Eq. (2) can sort out the proper teacher models and why these teacher models can enhance the performance of the student model.
> >
> > Additionally, I saw a very related work, FedBoost [1], that is neither discussed in the paper nor mentioned in the rebuttal. I also noticed that the authors have not addressed other reviewers' comments. Therefore, I will not give a better rating for this work.
> >
> > **Reference:**
> > [1] FedBoost: A Communication-Efficient Algorithm for Federated Learning

---

> > > ### Author Response · Authors · 2024-12-01
> > > **Response to follow-up questions (1/2)**
> > >
> > > We thank the reviewer for their follow-up questions and for the opportunity to further clarify these aspects of QKT.
> > >
> > > ### 1. Client Resource Constraints and Handling Multiple Teacher Models
> > > Thank you for highlighting the potential computational and memory challenges of handling multiple teacher models during training. As described in lines 210–219, the client selects only the most relevant teacher models for the query task using our filtering mechanism. Additionally, standard model offloading techniques can be employed, enabling clients to load and process teacher models sequentially rather than concurrently. More specifically, the teacher models are used only to extract the features for distillation; thus, these features can be extracted ahead of time sequentially. While this approach increases runtime slightly, it significantly reduces memory and computational overhead, a common practice in federated and distributed learning settings with resource constraints.
> > >
> > > QKT’s single-round knowledge transfer further optimizes resource consumption by eliminating the iterative multi-round communication cycles typical in traditional collaborative learning approaches, offering notable computational and communication savings.
> > >
> > > For highly resource-constrained settings, potential extensions such as compressing teacher models, employing lightweight surrogates, or leveraging federated pruning techniques could further alleviate computational and memory requirements. These strategies provide promising directions for future work to enhance QKT’s scalability and adaptability.
> > >
> > > We hope this explanation clarifies how QKT balances computational efficiency with collaborative learning benefits, addressing concerns about client resource constraints.
> > >
> > > ### 2. Justification of Equation (2)
> > > We appreciate your observation regarding Equation (2) and welcome the opportunity to clarify its theoretical basis and empirical validation.
> > >
> > > First, the assertion that outputs should be entirely random for unseen data points is not accurate. Neural networks are statistical models, meaning a well-trained model assigns probabilities based on its learned statistical distribution. For Gaussian noise inputs, the model does not produce completely random outputs but rather assigns probabilities reflecting its learned biases and overconfidence tendencies (Guo et al., NeurIPS 2017). Specifically, the overconfidence property ensures that even for out-of-distribution inputs, models assign disproportionately high probabilities to certain classes, creating distinct activation patterns.
> > >
> > > Second, Gaussian noise inputs act as approximations to average images across all classes due to the law of large numbers. Over many samples, Gaussian noise effectively provides a statistical average, allowing the model’s response to reflect its learned distribution. This mechanism enables QKT to identify relevant teacher models by analyzing their output probabilities on Gaussian noise.
> > >
> > > Empirical results validate this theoretical rationale. As shown in Figure 5, the predicted probabilities generated by Gaussian noise inputs closely align with the normalized actual data distribution of each teacher model, with low MSE values confirming the reliability of this method. By setting an appropriate threshold in Equation (2), QKT effectively filters irrelevant teachers, focusing on those with expertise relevant to the query classes.
> > >
> > > Finally, Section 4.3 (Table 3) demonstrates the practical benefits of filtering teachers using Equation (2). The results show significant improvements in query accuracy gain compared to naive KD, confirming that this filtering method isolates valuable expertise while minimizing noise and irrelevant contributions.
> > >
> > > We hope this clarification addresses your concerns and underscores the theoretical and empirical validity of using Equation (2) for teacher filtering in QKT.

---

> > > ### Author Response · Authors · 2024-12-01
> > > **Response to follow-up questions (2/2)**
> > >
> > > ### 3. Regarding FedBoost Baseline
> > > We appreciate your observation regarding FedBoost and its relevance to communication-efficient federated learning. While FedBoost optimizes a global ensemble model by transmitting a sampled subset of teacher models and ensemble weights in each round, this design inherently involves multiple communication rounds and requires sending several models to clients. This makes it more communication-intensive than traditional FL methods like FedAvg, which only transmit a single global model. By contrast, QKT completes knowledge transfer in a single communication round, significantly reducing communication overhead.
> > >
> > > Moreover, FedBoost primarily focuses on achieving generalized performance across clients, without explicitly addressing the challenges of data heterogeneity or supporting client-specific objectives. In heterogeneous environments, ensemble-based methods—shown in our experiments (Table 2)—struggle to adapt to individual client needs. Similarly, even more advanced FL approaches like FedProx and MOON fall short when addressing query-based tasks in such settings. In contrast, QKT employs teacher filtering and data-free masking to efficiently transfer targeted knowledge tailored to each client, achieving superior performance under heterogeneous conditions.
> > >
> > > While we acknowledge the strengths of FedBoost, particularly its theoretical contributions to ensemble-based FL, it appears better suited for more homogeneous or generalizable tasks.
> > >
> > >
> > > ### 4. Rebuttal Timeline and Review Process
> > > We are actively addressing all reviewer comments, including conducting additional experiments. The discussion period remains ongoing, and we aim to provide comprehensive responses by the deadline.

---

> > > > ### Author Response · Authors · 2024-12-02
> > > > **Update on FedBoost Evaluation**
> > > >
> > > > We wanted to inform you that, based on your suggestion, we have been actively evaluating FedBoost. The experiments are nearing completion, and we plan to post the results shortly.

---

> ### Author Response · Authors · 2024-12-02
> **Results of FedBoost Evaluation**
>
> We have evaluated FedBoost using both uniform and weighted sampling strategies for pre-trained base predictors under different configurations of communication budgets (\(C=5\) and \(C=10\)) using CIFAR-10 with pathological data distributions for single-class and multi-class queries. These experiments followed the setup detailed in Section 4, with additional baselines and results presented in Appendix A.7.
>
> The table below summarizes the results. Across all configurations, QKT consistently outperforms FedBoost on accuracy, query accuracy gain, and forgetting metrics. Notably, FedBoost’s reliance on multiple communication rounds and global ensemble optimization limits its adaptability to heterogeneous data and client-specific objectives.
>
>
>
>
> | **Queries**       | **Method**                              | **Acc**   | **Query Acc. Gain** | **Forgetting** |
> |-----------------------|------------------------------------------|-----------|----------------------|-----------------|
> | **Single-class queries** | FedBoost ($\gamma_k = \text{weighted}, C=5$) | 0.2362    | 0.2080              | -0.5711        |
> |                       | FedBoost ($\gamma_k = \text{weighted}, C=10$)| 0.2304    | 0.1790              | -0.6154        |
> |                       | FedBoost ($\gamma_k = \text{uniform}, C=5$)  | 0.0850    | 0.0080              | -0.7260        |
> |                       | FedBoost ($\gamma_k = \text{uniform}, C=10$) | 0.2066    | 0.1170              | -0.5915        |
> |                       | QKT                                       | 0.7456    | 0.7797              | -0.1681        |
> |-----------------------|------------------------------------------|-----------|----------------------|-----------------|
> | **Multi-class queries** | FedBoost ($\gamma_k = \text{weighted}, C=5$) | 0.2616    | 0.2419              | -0.5711        |
> |                       | FedBoost ($\gamma_k = \text{weighted}, C=10$)| 0.2893    | 0.3137              | -0.6155        |
> |                       | FedBoost ($\gamma_k = \text{uniform}, C=5$)  | 0.1494    | 0.1305              | -0.7260        |
> |                       | FedBoost ($\gamma_k = \text{uniform}, C=10$) | 0.2722    | 0.2565              | -0.5915        |
> |                       | QKT                                       | 0.6560    | 0.7140              | -0.3263        |
>
>
>
>
> We thank the reviewer for suggesting this comparison and hope these results clarify QKT’s advantages in decentralized and heterogeneous learning scenarios.

---

> > ### Comment · Reviewer_mLpA · 2024-12-02
> >
> > Thanks for your response. I think the response has partially addressed my concerns. Considering that you have put significant efforts in the rebuttal, I decide to raise the rating.

---

### Official Review · Reviewer_ycUP · 2024-11-10

**Soundness:** 3
**Presentation:** 2
**Contribution:** 3
**Rating:** 6
**Confidence:** 2

**Summary:**

The authors propose the Query-based Knowledge Transfer (QKT) method where the learning process consists of two stages: 1) the feature extractor enhancement stage, where the model is optimized to approximate the prediction of the teacher model regarding the query classes, and 2) the classification head refinement stage, where the initialized classification head is reused and trained to maintain the accuracy of the original tasks. The method is empirically shown to be effective regarding the query-class accuracy gain and the local-class forgetting.

**Strengths:**

Originality: Although knowledge distillation is well-known, the authors additionally propose a masking strategy for query-oriented learning, and add a second stage in which they reuse the classification head to maintain performance.

Significance: The method is empirically shown to be effective both in improving the query-class performance and maintaining the local-class performance.

Strengths:
1. The masking strategy is innovative and motivated based on the observation made by (Guo et al 2017).
2. The experiments are comprehensive with multiple popular benchmarks included. Experimental results show strong improvement across datasets, even with the light version of QKT.
3. The effects of various factors are presented.

**Weaknesses:**

1. It is unclear what Figure 2 is showing. The background color is inconsistent. Besides, it is not obvious that the QKT improves the boundaries.
2. As the losses in both stages are both equation 4, it is not clear why the second stage helps to maintain the performance. Could the author provide some intuitions behind the strategy?
3. While the performances of different methods are reported, the time of convergence and the computational costs in communication are missing. It would be beneficial for authors to show the speed and computational costs of QKT compared to the baselines. The convergence can be slow due to the synthetic dataset generation and query-class identification of dynamic models. The datasets shown are mostly small toy datasets. It is unclear whether the method scales.
4. There is no experiment showing how the performance changes during stages 1 and 2. It could be insightful for the authors to show the dynamic performance within the stages.
5. The performance is evaluated with 10 clients. Could the author show how the baselines and the QKT method perform with an increasing amount of clients?

**Questions:**

As listed in the weakness.

---

> ### Author Response · Authors · 2024-11-25
> **Response 1/n**
>
> Thank you for your valuable comments and suggestions. We're glad that you found our approach innovative, particularly our masking strategy, and that our comprehensive experiments across popular benchmarks effectively demonstrated strong improvements. Here is our responses to each point.
>
> ### **1. It is unclear what Figure 2 is showing.**
> Thank you for your feedback. We appreciate the opportunity to clarify and improve the Figure presentation.
> The background color inconsistencies were unintended and may have contributed to confusion. We have revised the plot to ensure consistent background colors, enhancing clarity and focus on the decision boundaries.
>
> The improvements in decision boundaries achieved by QKT are evident in the progression of the plots. As mentioned in lines 184-187, QKT Phase 1 introduces more structured and accurate boundaries for the query class compared to Naive KD. Phase 2 further refines these boundaries by focusing on the classification head, resulting in better separation and alignment for both query and local classes. This improvement is supported by the accuracy gains observed for certain classes, as discussed in lines 180–187 and 194.
>
> ### **2. it is not clear why the second stage helps to maintain the performance**
> Thank you for this insightful question. We hope to clarify that the intuition behind the stage is detailed in section 3.1 (Critical Role of Task-Related Parameters) and Figure 2. Here, we re-explain the intuition.  The purpose of our two-stage approach, despite both stages using Equation 4, is fundamentally tied to the targeted refinement in each phase.
>
> In the first stage, we focus on enhancing the feature extractor by transferring relevant query-specific knowledge with masking techniques that filter out irrelevant knowledge. However, this query-focused learning can potentially introduce interference with local class representations.
>
> To address this, we begin Stage 2 by replacing the classification head, which helps to retain the original local knowledge by restoring decision boundaries learned for local classes. Then, with the feature extractor frozen, we fine-tune only the classification head to incorporate the query knowledge. This approach stabilizes the knowledge captured in the feature extractor during Phase 1, allowing us to focus adjustments solely within the head. This selective refinement helps the model retain prior local knowledge while effectively learning the query classes.
>
> This strategy supports robust performance on both query and original classes, and we hope it clarifies the intuition behind our approach.
>
> ### **3. speed and computational costs of QKT compared to the baselines**
> Thank you for this insightful comment. We would like to clarify that the synthetic dataset generation and query-class identification in QKT are highly computationally efficient. Specifically, only a small number of Gaussian noise samples (20 in our experiments, line 443) are generated, with no additional computation or optimization required. This is followed by a straightforward inference step to collect the probability distribution and detect relevant models for the query task, minimizing both time and computational overhead.
>
> In terms of communication, QKT is designed for a single-round transfer, contrasting with multi-round federated learning approaches that often require iterative updates and communication. We quantify communication by the number of rounds required (line 361), and report this metric in Table 2. This gives QKT an advantage in terms of communication efficiency, particularly when network resources are limited.
>
> Regarding dataset size, we acknowledge the value of experiments on larger datasets to further demonstrate scalability. However, we hope to clarify that our experimental setting closely follows the prior work to ensure fair comparisons (e.g., McMahan et al. (AISTATS 2017), Li et al. (MLSys 2020), Fallah et al. (AAAI 2021), and Deng et al. (ICLR 2022)). It is also a common practice, as we do, to apply various heterogeneous distributions on the datasets we use to show the potential scalability and performance of the algorithm.
>
> We hope this explanation provides clarity on QKT’s computational efficiency and scalability.

---

> ### Author Response · Authors · 2024-11-25
> **Response 2/n**
>
> ### **4. how the performance changes during stages 1 and 2**
> Thank you for your insightful comment regarding the dynamic performance across the two phases of QKT. We have an ablation study to show the performance of the two stages in QKT (lines 413-418, table 3). We further include an attached plot (Appendix A.5.1 in revised manuscript) that compares the validation accuracy after each epoch during Phase 1 and Phase 2 for a random subset of clients trained on CIFAR-10, alongside the initial local accuracy (orange) before Phase 1. The results show that
> - Both Phase 1 and Phase 2 of QKT show significant improvements (more than 80%), far surpassing the local accuracy baseline (less than 45%).
> - Phase 2 demonstrates remarkable stability compared to Phase 1, as freezing the feature extractor preserves its learned representations. By focusing exclusively on classification head refinement, Phase 2 achieves smoother and more consistent accuracy improvements, avoiding the fluctuations seen in Phase 1 caused by simultaneous updates to both components.
>
> This two-phase design effectively balances adaptability and stability, addressing the challenges of heterogeneous collaborative learning environments.

---

> ### Author Response · Authors · 2024-11-25
> **Response 3/n**
>
> ### **5.  baselines and the QKT method perform with an increasing amount of clients**
> Thank you for your request. In response, we expanded our evaluation to include 50 clients using CIFAR-10 under a pathological distribution, building on the experimental setup described in Section 4.1 of the paper. We followed the same evaluation metrics outlined in lines 355–366 and Appendix A.1 to ensure consistency with our main experiments.
>
> We selected the main baselines that achieved competitive results in our earlier experiments. To account for the increased number of clients,  we adopted typical participation rates (P%) of 10% (Table 1) and 20% (Table 2), as commonly used in cross-device federated learning. For FL methods, this represents the number of clients selected for each round, while for single-round methods like QKT, it represents the number of clients participating in the single knowledge transfer round.  Additionally, we included the "Ensemble (all models)" baseline to assess performance when utilizing all client models simultaneously.
>
> At a larger scale, with 50 clients, QKT consistently outperformed other baselines, achieving higher accuracy, greater query accuracy gain, and lower forgetting rates. It is worth noting that QKT is highly efficient in that it completes knowledge transfer in a single round, significantly reducing communication, storage, and computational demands.
>
>
> Table1: Performance of QKT and baselines with 50 clients (CIFAR-10, pathological distribution) at a 10% participation rate.
>
> |        P%: 0.1       |                       |    Acc.   | Query Acc. gain | Forgetting | Uniform Acc. |
> |:--------------------:|-----------------------|:---------:|:---------------:|:----------:|:------------:|
> | Single class queries |         FedAvg        |  0.355533 |     0.346319    |  -0.421399 |   0.3601400  |
> |                      | Ensemble (all models) |  0.377580 |     0.391380    |  -0.42033  |   0.370679   |
> |                      |        Ensemble       |  0.224559 |     0.238380    |  -0.580839 |   0.217650   |
> |                      |           KD          |  0.321850 |     0.158440    |  -0.291293 |    0.40355   |
> |                      |          QKT          |  0.590206 |     0.512379    |  -0.122973 |   0.629119   |
> |  Multi-class queries |         FedAvg        |  0.372590 |     0.377289    |  -0.421399 |   0.3693679  |
> |                      | Ensemble (all models) |  0.361797 |     0.348241    |  -0.420339 |   0.366202   |
> |                      |        Ensemble       | 0.2205406 |     0.224228    |  -0.558339 |   0.217794   |
> |                      |           KD          |  0.270026 |     0.180399    |  -0.291293 |   0.340036   |
> |                      |          QKT          |  0.442319 |     0.480412    |  -0.318020 |   0.430587   |
>
>
>
> Table2: Performance of QKT and baselines with 50 clients (CIFAR-10, pathological distribution) at a 20% participation rate.
> |        P%: 0.2       |                       |   Acc.   | Query Acc. gain | Forgetting | Uniform Acc. |
> |:--------------------:|-----------------------|:--------:|:---------------:|:----------:|:------------:|
> | Single class queries |         FedAvg        | 0.325116 |     0.304820    |  -0.439373 |   0.335264   |
> |                      | Ensemble (all models) | 0.377580 |     0.391380    |  -0.42033  |   0.3706799  |
> |                      |        Ensemble       |  0.28953 |     0.31402     |  -0.524606 |   0.277285   |
> |                      |           KD          |  0.33096 |     0.235680    |  -0.346913 |   0.378609   |
> |                      |          QKT          |  0.64558 |     0.652679    |  -0.144859 |   0.642039   |
> |  Multi-class queries |         FedAvg        | 0.379309 |     0.399826    |  -0.43937  |   0.365998   |
> |                      | Ensemble (all models) | 0.361797 |     0.348241    |  -0.420339 |   0.366202   |
> |                      |        Ensemble       | 0.255061 |     0.252811    |  -0.548939 |   0.254010   |
> |                      |           KD          |  0.28853 |     0.252986    |  -0.395126 |   0.316938   |
> |                      |          QKT          | 0.442734 |     0.488701    |  -0.410544 |   0.409805   |
>
> We appreciate this opportunity to demonstrate QKT’s effectiveness in handling larger-scale collaborative learning scenarios.

---

> ### Author Response · Authors · 2024-12-02
> **Follow-Up to Reviewer ycUP**
>
> We wanted to express our gratitude once again to the reviewer for their valuable feedback and for highlighting key aspects of our work. We hope that the detailed responses and additional experiments we provided have clarified the points you raised, including the revised Figure 2, the intuition behind our two-stage approach, the computational efficiency and scalability of QKT, and the dynamic performance across stages.
>
> Additionally, we expanded our evaluation to include experiments with 50 clients to further demonstrate QKT’s scalability. These results consistently show that QKT achieves superior performance compared to the baselines across various metrics.
>
> If there are any remaining concerns or additional clarifications needed, we’d be more than happy to address them before the discussion period concludes. We also kindly invite you to reconsider your score in light of the additional evidence and improvements we’ve presented.
>
> Thank you for taking the time to review our work and for helping us improve it.

---

### Meta-Review · Area_Chair_4LJq · 2024-12-20

**Metareview:**

The paper studies knowledge transfer under heterogeneity.  Different clients train local models with their local data, and knowledge is distilled across clients using pretrained models as teachers. Irrelevant teacher models are filtered out via masking in distillation (without data exchange, through an appropriate query submission) and the head of the student is refined to avoid catastrophic forgetting. Experiments show good performance w.r.t. predictive power in new tasks and forgetting.

Reviewers raised several concerns regarding scaling with the number of clients, as well as w.r.t. comparison to other competitors, the impact of heterogeneity, and many more, that the authors have addressed during the rebuttal stage.

**Additional Comments On Reviewer Discussion:**

A lingering issue that remained was that communication complexity was only discussed in terms of rounds. The authors are encouraged to elaborate on the amounts of bits required to be transferred.

---

### Decision · Program_Chairs · 2025-01-22

Accept (Poster)